# GENERALIZABLE MONOCULAR 3D HUMAN RENDERING VIA DIRECT GAUSSIAN ATTRIBUTE DIFFUSION

## ABSTRACT

This paper leverages 3D Gaussian Splatting to tackle the challenging task of generating novel views of humans from given single-view images. Existing methods typically adopt an *indirect* supervision manner, i.e., splat-based rasterization for differentiable rendering. However, the intricate coupling of various 3D Gaussian attributes complicates precise error backpropagation during optimization, often resulting in convergence to local optima. In contrast, we propose a novel *direct paradigm* to train a conditional diffusion model directly supervised by proxy-ground-truth 3D Gaussian attributes. Specifically, we propose a two-stage construction process to derive consistent and smoothly distributed proxy-ground-truth 3D Gaussian attributes. Subsequently, we train a point-based conditional diffusion model customized to learn the data distribution of these proxy attributes. The resulting diffusion model can generate the 3D Gaussian attributes for the input single-view image, which are further rendered into novel views. Extensive experimental results showcase the significant performance advancement of our method over state-of-the-art approaches. Source code will be made publicly available.

## 1 INTRODUCTION

Image-driven 3D human digitization has gained significant attention due to its rich applications in film-making, game production, AR/VR, immersive telepresence. In recent years, there has emerged a series of works performing novel view synthesis (NVS) from observed images of human captures.

**Image**-based approaches (Saito et al., 2019; Zhang et al., 2023; Ho et al., 2024; Zhang et al., 2024) infer the geometric surface of the human body and then perform texture estimation, which typically suffers from relatively lower resolution. **Video/multi-view**-based approaches (Mildenhall et al., 2020; Gao et al., 2022; Weng et al., 2022; Hu et al., 2024) can produce more fine-grained visual appearance, but require multi-view images or monocular videos as inputs. SHERF (Hu et al., 2023) makes the first attempt to build a single-view generalizable NeRF learning framework, but still lacks the ability to restore accurate geometric details. More recently, 3DGS (Kerbl et al., 2023) rapidly evolves, serving as a more powerful neural rendering pipeline, suggesting another promising way (Zheng et al., 2024) of achieving real-time human NVS.

The most common way of building generalizable 3DGS (Zheng et al., 2024; Zou et al., 2024) is to train a parameterized neural model for generating the desired Gaussian attribute set, and then impose supervision by comparing pixel-level differences between the rendered and observed images (see Fig. 1(a)). **However, the supervision gap between the pixel and attribute domains, as well as the many-to-one mapping relationship between the Gaussian attribute set and rendered image cast doubt on the training effectiveness.** To address these issues, we propose to directly impose attribute-level supervision by pre-creating proxy-ground-truth Gaussian attribute sets by customizing a two-stage workflow, including per-scene overfitting and distribution unification.

By leveraging proxy-ground-truth Gaussian attributes, we establish a *direct paradigm* utilizing an attribute-wise loss function for supervision during training (see Fig. 1(b)). We adopt a diffusion-based framework to tackle this challenge as a conditional generation task, focusing on modeling the distribution of 3D Gaussian attributes conditioned on a monocular image. While existing point-based diffusion models are tailored for tasks like point cloud generation, completion, or upsampling (Luo & Hu, 2021; Zhou et al., 2021; Qu et al., 2024), they are not directly applicable to our

Figure 1: HUMAN-DAD adopts a novel training paradigm. Distinct from the traditional paradigm, which requires a splat-based rasterizer for *indirect* supervision, HUMAN-DAD *directly* utilizes proxy-ground-truth 3D Gaussian attributes to learn a conditional 3D Gaussian attribute diffusion model.

specific task of generating 3D Gaussian attributes. To overcome this hurdle, we devise a conditional 3D **Human** Gaussian diffusion framework via **D**irect Gaussian **A**ttribute **D**iffusion, named HUMAN-DAD. Human-DAD comprises two core modules. The initial module leverages a human reconstruction pipeline to generate point clouds, which are then initialized as 3D Gaussian positions. In the second module, we propose a conditional Gaussian attribute diffusion module to learn the data distribution of 3D Gaussian attributes, enabling the generation of realistic and plausible results. Moreover, we have thoughtfully crafted the input conditions, consisting of pixel-aligned features and SMPL-semantic features.

In summary, the main contributions of this work are:

- we demonstrate the inefficacy of the traditional *indirect paradigm* and propose a two-stage construction method to attain proxy ground truth 3D Gaussian attributes, thereby enabling a *direct paradigm* for attribute-wise supervision. This ultimately enhances the effectiveness of the training process.
- we design a novel generalizable 3DGS framework HUMAN-DAD and design a conditional Gaussian diffusion module for learning the monocular image conditioned distribution of 3D Gaussian attributes, which also exhibits the potential of diffusion models for other applications in the field of 3DGS; and
- we have conducted extensive and comprehensive experiments to demonstrate the rationality of our proxy-ground-truth attribute sets construction process and to validate the effectiveness of HUMAN-DAD both quantitatively and qualitatively. Moreover, our HUMAN-DAD achieves state-of-the-art performance.

## 2 RELATED WORK

### 2.1 HUMAN RECONSTRUCTION

**Geometry Reconstruction.** There exist various 3D scene representations used in different applications, such as voxels, point clouds, meshes, and implicit functions. Currently, implicit functions have been widely applied to recovering the geometric surfaces of the human body. Saito et al. (2019; 2020) pioneeringly employed implicit functions for monocular human body reconstruction by regressing the occupancy value of each point in space and then using the marching cubes algorithm to extract the human body surface. However, due to relying solely on image features and lacking human body priors, they often struggled with occluded humans. To address this issue, PAMIR and ICON (Zheng et al., 2021; Xiu et al., 2022) introduce the parametric human model SMPL to make implicit functions aware of human body structure. SiFU (Zhang et al., 2024) employs a text-to-image diffusion model to predict invisible information and generate realistic results. However, the strong constraints of SMPL make implicit-based methods heavily reliant on the accuracy of SMPL estimation, and they typically require numerous query points to extract 3D surfaces. In comparison, HaP (Tang et al., 2023), a purely explicit-based method, offers greater flexibility in manipulating the human body as a point cloud in 3D space, enabling unconstrained-topology modeling of arbitrarily-clothed human body shapes. Owing to its efficiency and effectiveness, we utilize HaP to generate the 3D Gaussian positions in this paper.

**Novel View Synthesis.** Recently, an implicit neural radiance representation (NeRF) (Mildenhall et al., 2020) has achieved great success in synthesizing high-quality novel views. By equipping the parametric human model SMPL, NeRF-based (Jiang et al., 2022; Xu et al., 2021) methods do not re-

quire ground truth 3D geometry information and are capable of rendering such high-quality views of humans. NeuralBody (Peng et al., 2021) innovatively adopts NeRF for human novel view synthesis by learning the structured latent code of the canonical SMPL model among different frames. HumanNerf (Weng et al., 2022) decomposes the rigid skeleton motion and non-rigid clothes motion in monocular videos to learn a better NeRF representation in the canonical space. More recently, some 3DGS-based methods (Hu et al., 2024; Kocabas et al., 2024) have attempted to learn 3D Gaussian attributes in the canonical space for fast rendering speed. However, these methods typically require monocular or multi-view videos as input for per-scene optimization and cannot be generalized to other scenes.

## 2.2 GENERALIZABLE NOVEL VIEW SYNTHESIS

Large language models have achieved remarkable success in text-to-image generation. While several methods (Liu et al., 2023; Yang et al., 2024; Xue et al., 2024) attempted to exploit this strong imaginative ability to generate novel views of a single image with text prompts, they face difficulties in maintaining consistency among different views due to the absence of 3D information for supervision. Zhao et al. (2022) and Gao et al. (2022) projected the SMPL models to multi-view images and combine the image features with the canonical SMPL features to implement generalizable human NeRFs. SHERF Hu et al. (2023) further proposed a hierarchical feature map to realize a generalizable single-view human NeRF. More recently, several works attempt to train generalizable 3DGS models (Liu et al., 2024; Tang et al., 2024; Zou et al., 2024; Zheng et al., 2024; Wang et al., 2024). (Zou et al., 2024) combines the triplane representation and the 3DGS representation, following PIFu to **prepare the pixel-aligned feature for** training a generalizable 3DGS model. Zheng et al. (2024) designed a generalizable multi-view human novel view synthesis framework by composing an iterative depth estimation module and a Gaussian parameter regression module. These generalizable 3DGS methods follow the *indirect paradigm*. *In contrast*, we introduce a *direct paradigm* that supervises the training process with ground truth 3D Gaussian attributes to provide attribute-wise loss.

# 3 PROPOSED METHOD

## 3.1 PRELIMINARY OF 3DGS

Different from implicit neural representation approaches (Mildenhall et al., 2020; Park et al., 2019), 3DGS (Kerbl et al., 2023) explicitly encodes a radiance field as an unordered set of Gaussian primitives denoted as $\mathcal{A} = \{\mathbf{a}^{(n)}\}_{n=1}^N$. Each primitive is associated with the set of optimizable attributes:

$$\mathbf{a}^{(n)} = \{\mathbf{p}^{(n)}, \alpha^{(n)}, \mathbf{s}^{(n)}, \mathbf{q}^{(n)}, \mathbf{c}^{(n)}\}, \tag{1}$$

including position $\mathbf{p}^{(n)} \in \mathbb{R}^3$, opacity value $\alpha^{(n)} \in \mathbb{R}$, scaling factor $\mathbf{s}^{(n)} \in \mathbb{R}^3$, rotation quaternion $\mathbf{q}^{(n)} \in \mathbb{R}^4$, and spherical harmonics (SH) coefficients $\mathbf{c}^{(n)} \in \mathbb{R}^d$. For an arbitrary viewpoint with camera parameters $\mathcal{V}$, a differentiable tile rasterizer $\mathcal{R}$ is applied to render the Gaussian attribute set $\mathcal{A}$ into the corresponding view image $\mathbf{I}_r$, which can be formulated as:

$$\mathbf{I}_r = \mathcal{R}(\mathcal{A}; \mathcal{V}). \tag{2}$$

For a set of observed $K$ multi-view images $\{\mathbf{I}^{(k)}\}_{k=1}^K$ depicting a specific scene, together with their calibrated camera parameters $\{\mathcal{V}^{(k)}\}_{k=1}^K$, the optimization process iteratively updates the Gaussian attributes by comparing the difference between rendered images and observed ground-truths, which can be formulated as:

$$\mathbf{I}_r^{(k)} = \mathcal{R}(\mathcal{A}; \mathcal{V}^{(k)}), \quad \min_{\mathcal{A}} \sum_{k=1}^K \ell_{\mathrm{pmet}}(\mathbf{I}_r^{(k)}, \mathbf{I}^{(k)}), \tag{3}$$

where $\ell_{\mathrm{pmet}}(\cdot, \cdot)$ computes the pixel-wise photometric error within the image domain. After training, the resulting optimized Gaussian attribute set $\mathcal{A}$ serves as a high-accuracy neural representation of the target scene for real-time NVS. However, despite the fast inference speed of 3DGS, scene-specific overfitting still requires at least several minutes to complete.

Figure 2: The two-stage workflow of creating proxy-ground-truth Gaussian attributes.

## 3.2 TRAINING PARADIGM OF GENERALIZABLE FEED-FORWARD 3DGS

In contrast to the conventional working mode of per-scene overfitting, many recent studies are devoted to constructing generalizable 3DGS frameworks (Liu et al., 2024; Tang et al., 2024; Zou et al., 2024; Zheng et al., 2024; Wang et al., 2024) via shifting the actual optimization target from the Gaussian attribute set to a separately parameterized learning model $\mathcal{M}(*; \Theta)$, where $*$ denotes network inputs and $\Theta$ denotes network parameters. Generally, we can summarize that all such approaches uniformly share the same training paradigm, where the learning model $\mathcal{M}(\cdot, \cdot)$ consumes its input to generate scene-specific Gaussian attributes at the output end. Through differentiable rendering $\mathcal{R}$, these approaches impose ***pixel-level supervision*** in the image domain, as formulated below:

$$\min_{\Theta} \sum_{k=1}^{K} \ell_{\mathrm{pmet}}(\mathcal{R}(\mathcal{M}(*; \Theta); \mathcal{V}^{(k)}), \mathbf{I}^{(k)}). \tag{4}$$

Though such a training paradigm is reasonable and straightforward, the learning effectiveness can be weakened due to the gap between the image domain where supervision is imposed and the actually desired Gaussian attribute domain. **More importantly, different Gaussian attribute sets can produce the same rendered image. Such a many-to-one mapping relationship can complicate the solution space and cause certain degrees of confusion for model learning.**

The above-analyzed drawbacks of the existing mainstream training paradigm naturally motivate us to get rid of what we call the ***supervision gap***. Accordingly, we propose a paradigm shift by directly imposing ***attribute-level supervision***. Under our targeted setting with single-view image $\mathbf{I}_s$ as input, the proposed training paradigm can be formulated as:

$$\mathcal{A} = \mathcal{M}(\mathbf{I}_s; \Theta), \quad \min_{\Theta} \ell_{\mathrm{setdiff}}(\mathcal{A}, \hat{\mathcal{A}}), \tag{5}$$

where $\ell_{\mathrm{setdiff}}(\cdot, \cdot)$ measures the primitive difference between the predicted Gaussian attribute set $\mathcal{A}$ and the pre-created proxy-ground-truth attribute set $\hat{\mathcal{A}}$.

Overall, our proposed single-view generalizable human 3DGS learning framework consists of two core processing phases: 1) *creating proxy-ground-truth Gaussian attributes as supervision signals*, and 2) *training a conditional diffusion model for Gaussian attribute generation*, as introduced in the following Sections 3.3 and 3.4.

## 3.3 CREATION OF PROXY-GROUND-TRUTH 3D GAUSSIAN ATTRIBUTE SETS

To facilitate direct attribute-level optimization, we need to pre-create a dataset of proxy-ground-truth Gaussian attribute sets serving as the actual supervision signals for training $\mathcal{M}$. Formally, suppose that our raw training dataset is composed of totally $J$ different human captures each associated with multi-view image observations $\{\mathbf{I}_j^{(k)}\}_{k=1}^{K}$ and camera parameters $\{\mathcal{V}_j^{(k)}\}_{k=1}^{K}$, we aim to produce the corresponding proxy-ground-truths $\{\hat{\mathcal{A}}\}_{j=1}^{J}$ as:

$$\hat{\mathcal{A}}_j = \{\hat{\mathbf{a}}_j^{(n)}\}_{n=1}^{N} = \{\hat{\mathbf{p}}_j^{(n)}, \hat{\alpha}_j^{(n)}, \hat{\mathbf{s}}_j^{(n)}, \hat{\mathbf{q}}_j^{(n)}, \hat{\mathbf{c}}_j^{(n)}\}. \tag{6}$$

In fact, the most straightforward way of obtaining $\{\hat{\mathcal{A}}_j\}_{j=1}^{J}$ is to separately overfit the vanilla 3DGS over each of the $J$ human captures and save the resulting Gaussian attributes. Unfortunately, owing to the inevitable randomness of gradient-based optimization and primitive manipulation, the overall distributions of the independently optimized Gaussian attribute sets are typically inconsistent. Even for the exactly same scene, two different runs produce varying Gaussian attribute sets (e.g., primitive density and orders, attribute values), which results in a chaotic and hard-to-learn solution space.

Figure 3: The flowchart of our HUMAN-DAD. HUMAN-DAD predicts 3D Gaussian positions and a back-view image using a position generator and a stable diffusion module. It assigns SMPL semantic labels to points in 3D Gaussians, deducing an SMPL-semantic feature. The 3D Gaussians are decomposed for front- and back-view projection, achieving a pixel-aligned feature. Both features condition the 3DGS diffuser.

To obtain consistently-distributed Gaussian attribute sets for shrinking the solution space, we particularly develop a two-stage proxy-ground-truth creation workflow as depicted in Fig. 2. Considering our task characteristics, we uniformly sample a dense 3D point cloud from the ground-truth human body surface to serve as the desired Gaussian positions $\{\hat{\mathbf{p}}_j^{(n)}\}_{n=1}^N$. The other four types of attributes (i.e., opacities, scalings, rotations, SHs) are deduced from two sequential processing stages of what we call *per-scene overfitting* and *distribution unification*, as introduced below.

**Stage 1: Per-Scene Overfitting.** This stage independently performs 3DGS overfitting over each of the $J$ human captures, but with one subtle difference from the vanilla optimization scheme. Specifically, instead of directly maintaining Gaussian attributes as learnable variables, we introduce a point cloud learning network $\mathcal{F}_1(\cdot; \Phi_1)$, which consumes $\{\hat{\mathbf{p}}_j^{(n)}\}_{n=1}^N$ at the input end and outputs the rest types of Gaussian attributes, as formulated below:

$$\{\bar{\alpha}_j^{(n)}, \bar{\mathbf{s}}_j^{(n)}, \bar{\mathbf{q}}_j^{(n)}, \bar{\mathbf{c}}_j^{(n)}\}_{n=1}^N = \mathcal{F}_1(\{\hat{\mathbf{p}}_j^{(n)}\}_{n=1}^N; \Phi_1). \tag{7}$$

The purpose of moving the Gaussian attributes to the output end of a neural network lies in exploiting the inherent smoothness tendency (Rahaman et al., 2019) of neural network's outputs. Accordingly, for the $j$-th training sample, the per-scene optimization objective can be formulated as:

$$\bar{\mathcal{A}}_j = \{\hat{\mathbf{p}}_j^{(n)}, \bar{\alpha}_j^{(n)}, \bar{\mathbf{s}}_j^{(n)}, \bar{\mathbf{q}}_j^{(n)}, \bar{\mathbf{c}}_j^{(n)}\}_{n=1}^N, \quad \min_{\Phi_1} \sum_{k=1}^K \ell_{\text{pmet}}(\mathcal{R}(\bar{\mathcal{A}}_j; \mathcal{V}_j^{(k)}); \mathbf{I}_j^{(k)}), \tag{8}$$

where $\ell_{\text{pmet}}$ involves both $L_1$ and SSIM measurements. Besides, auxiliary constraints are imposed over scaling and opacity attributes to suppress highly non-uniform distributions.

**Stage 2: Distribution Unification.** Though the preceding stage has preliminarily deduced a dataset of Gaussian attribute sets $\{\bar{\mathcal{A}}_j\}_{j=1}^J$, the per-scene independent optimization can still lead to certain degrees of randomness and distribution inconsistency. Therefore, in the second stage, we introduce another deep set architecture $\mathcal{F}_2(\cdot; \Phi_2)$ to overfit the whole $J$ training samples:

$$\{\hat{\mathcal{A}}_j\}_{j=1}^J = \mathcal{F}_2(\{\bar{\mathcal{A}}_j\}_{j=1}^J; \Phi_2), \quad \min_{\Phi_2} \sum_{j=1}^J \sum_{k=1}^K \ell_{\text{pmet}}(\mathcal{R}(\hat{\mathcal{A}}_j; \mathcal{V}_j^{(k)}); \mathbf{I}_j^{(k)}). \tag{9}$$

Since it is impractical to feed the whole $J$ training samples all at once, we adopt a batch-wise scheme with a certain number of training epochs, after which the resulting optimized $\{\hat{\mathcal{A}}_j\}_{j=1}^J$ serve as our required proxy-ground-truths.

Due to space limitation, the detailed network structures of $\mathcal{F}_1$ and $\mathcal{F}_2$ are presented in *Appendix* A.1. The functionality and necessity of the two processing stages are evaluated in experiments.

### 3.4 DIRECT GAUSSIAN ATTRIBUTE DIFFUSION

Having created a collection of proxy-ground-truth Gaussian attributes $\{\hat{\mathcal{A}}_j\}_{j=1}^J$ as supervision signals, we shift attention to modeling the target distribution $q(\mathcal{A}|\mathbf{I}_s)$ conditioned on the input single-view image $\mathbf{I}_s$ to predict the desired Gaussian attribute set $\mathcal{A}$. First, we tend to separately predict Gaussian positions, which is essentially a 3D point cloud. Second, we treat the obtained point cloud

as geometric priors and extract human-centered features, then feed them into a conditional diffusion pipeline for diffusing the rest types of Gaussian attributes.

**Generation of Gaussian Positions.** In the training phase of the generalizable human 3DGS framework, we directly use the Gaussian positions $\{\hat{\mathbf{p}}_j^{(n)}\}_{n=1}^N$ prepared in the proxy-ground-truth creation process. In the inference phase, we need to specifically estimate the Gaussian positions from the input image. In our implementation, we design a position generator with the rectification of the SMPL parametric human model.

The position generator begins with monocular depth estimation (Patni et al., 2024) to generate from $\mathbf{I}_s$ the corresponding depth map, which is converted into a partial 3D point cloud. In parallel, we also estimate from $\mathbf{I}_s$ the corresponding SMPL model, whose pose is further rectified by the partial point cloud. Then, we feed the rectified SMPL model and the partial point cloud into a point cloud generation network to output the desired set of 3D Gaussian positions. To promote the uniformity of point density, we perform point cloud upsampling and then apply farthest point sampling. In this way, we can stably obtain a set of accurate 3D Gaussian positions $\{\mathbf{p}^{(n)}\}_{n=1}^N$ as geometric priors.

**Extraction of Human-Centered Features.** To supplement more informative conditioning signals for the subsequent attribute diffusion, we further extract two aspects of human-centered features.

The first is *pixel-aligned features* for providing visual appearance information. To achieve this, we project the 3D Gaussian positions onto the input image space. Then we utilize (Brooks et al., 2023) to predict a back-view image $\mathbf{I}_{\text{back}}$ with respect to the view of $\mathbf{I}_s$. The visible and invisible partitions of $\{\mathbf{p}^{(n)}\}_{n=1}^N$ are respectively projected onto $\mathbf{I}_s$ and $\mathbf{I}_{\text{back}}$, and the feature maps corresponding to $\mathbf{I}_s$ and $\mathbf{I}_{\text{back}}$ are extracted via 2D CNNs. Finally, we concatenate the visible and invisible pixel-aligned features to form the pixel-aligned feature $\boldsymbol{\beta}^{(n)}$.

The second is *SMPL-semantic features* for strengthening the awareness of human body structure. To achieve this, we incorporate semantic labels defined on SMPL. We perform the nearest neighbor searching to identify the nearest points of the 3D Gaussians on the SMPL surface. For each 3D Gaussian, we retrieve the corresponding nearest SMPL point index, the distance, and the semantic label, which are embedded into the latent space through MLPs. The resulting feature embeddings are concatenated to assign each point the SMPL-semantic feature $\boldsymbol{\gamma}^{(n)}$.

**Conditional Diffusion.** Having obtained Gaussian positions $\mathbf{p}^{(n)}$, pixel-aligned features $\boldsymbol{\beta}^{(n)}$, and SMPL-semantic features $\boldsymbol{\gamma}^{(n)}$, we perform condition diffusion to generate the rest attributes including $\alpha^{(n)}$, $\mathbf{s}^{(n)}$, $\mathbf{q}^{(n)}$, and $\mathbf{c}^{(n)}$. Empirically, we observe that simultaneously diffusing all these four types of attributes usually results in training collapse. Through our exploration, we choose to separate the diffusion of SH coefficients and the other three types of attributes.

For training the generation of SH coefficients, we design an attribute diffuser $\text{GSDIFF}_{\psi_1}$ to predict the noise at the given time step $\mathbf{t}$ and use an $L_2$ loss for supervision:

$$\epsilon^{(n)} = \text{GSDIFF}_{\psi_1}(\tilde{\mathbf{c}}_t^{(n)}, \mathbf{p}^{(n)}, \boldsymbol{\beta}^{(n)}, \boldsymbol{\gamma}^{(n)}, \mathbf{t}), \quad \min_{\psi_1} \mathbb{E}_{\epsilon \sim \mathcal{N}} \|\hat{\epsilon}^{(n)} - \epsilon^{(n)}\|^2, \tag{10}$$

where $\tilde{\mathbf{c}}_t^{(n)}$ denote SH coefficients with noise added, and $\hat{\epsilon}^{(n)}$ is the ground-truth noise. For inference, we sample random SH coefficients $\mathbf{c}_T^{(n)}$ from the Gaussian distribution and iteratively remove noises to achieve $\mathbf{c}_0^{(n)}$. However, as demonstrated in PDR (Lyu et al., 2022), the inductive bias of the evidence lower bound (ELBO) is unclear in the 3D domain, resulting in $\mathbf{c}_0^{(n)}$ still containing noise, we further adopt an extra-step to remove the remained noises. Also, we predict the other attributes, i.e., $\alpha^{(n)}, \mathbf{s}^{(n)}, \mathbf{q}^{(n)}$ at this extra-step.

$$\{\epsilon^{(n)}, \alpha^{(n)}, \mathbf{s}^{(n)}, \mathbf{q}^{(n)}\} = \text{GSDIFF}_{\psi_2}(\mathbf{c}_0^{(n)}, \mathbf{p}^{(n)}, \boldsymbol{\beta}^{(n)}, \boldsymbol{\gamma}^{(n)}), \quad \mathbf{c}^{(n)} = \mathbf{c}_0^{(n)} - \epsilon^{(n)},$$
$$\min_{\psi_2} \|\mathbf{c}^{(n)} - \hat{\mathbf{c}}^{(n)}\|^2 + \|\alpha^{(n)} - \hat{\alpha}^{(n)}\|^2 + \|\mathbf{s}^{(n)} - \hat{\mathbf{s}}^{(n)}\|^2 + \|\mathbf{q}^{(n)} - \hat{\mathbf{q}}^{(n)}\|^2, \tag{11}$$

Finally, we obtain a 3D Gaussian attribute set $\{\mathbf{p}^{(n)}, \alpha^{(n)}, \mathbf{s}^{(n)}, \mathbf{q}^{(n)}, \mathbf{c}^{(n)}\}$ of a scene when given a single-view image $\mathbf{I}_s$, which can be used to render novel views of the human body. The details of the 3D Gaussian attribute diffusion model are in the *Appendix* A.2.

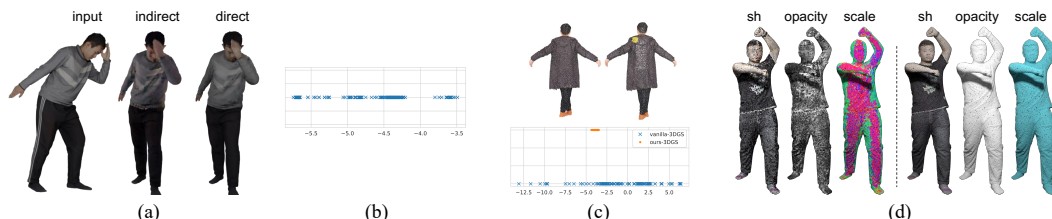

Figure 4: (a) Visual comparison between the indirect and direct paradigms. (b) The large variation of vanilla-3DGS spherical harmonic values after 200 attempts on a scene. (c) Spherical harmonic values comparison of the local area (marked with yellow) between vanilla-3DGS and our proxy-ground-truth 3D Gaussian attributes. (d) Visualization of spherical harmonic, opacity, and scale for vanilla-3DGS and our proxy-ground-truth 3D Gaussian attributes. 🔍 Zoom in for details.

## 4 EXPERIMENTS

### 4.1 DATASETS AND IMPLEMENTATION DETAILS

We employed 480 humans from Thuman2 (Yu et al., 2021) for the construction of proxy ground truth 3D Gaussian attributes and the training of the attribute diffusion model. And we quantitatively evaluated HUMAN-DAD on Thuman2 (20 humans), CityuHuman (20 humans) (Tang et al., 2023), 2K2K (25 humans) (Han et al., 2023) and CustomHuman (40 humans) (Ho et al., 2023). We adopted the peak signal-to-noise ratio (PSNR), structural similarity index (SSIM), and Learned Perceptual Image Patch Similarity (LPIPS) as evaluation metrics.

When constructing the proxy ground truth 3D Gaussian attributes, we rendered 360 views for each human, and we uniformly sampled 20000 points from each human surface, which was the initial 3D Gaussian position. In the first stage, we utilized a Point Transformer as the backbone to predict the 3D Gaussian attributes. For each human subject, we overfitted the Point Transformer for 4000 epochs, using the Adam optimizer with a learning rate of 0.0002. In the second stage, we employed another Point Transformer as the backbone. The batch size was set to 4, the number of epochs was set to 1300, and we continued to use the Adam optimizer with a learning rate of 0.0002. Other settings of 3DGS were following (Kerbl et al., 2023). To train HUMAN-DAD, we designed the attribute diffusion model using PointNet++ as the backbone[1]. ResNet18 with the pre-trained weights was used to extract the image feature when preparing the pixel-aligned feature. During training, the batch size was set to 4, the number of epochs was set to 300, the optimizer was Adam, and the learning rate was 0.0002.

### 4.2 ANALYSIS ON 3D GAUSSIAN ATTRIBUTE CONSTRUCTION

**Weakness of Indirect Paradigm.** We assert that the traditional *indirect paradigm* of utilizing a splat-based rasterizer for supervision is ineffective. To validate this claim, we devised two experimental settings. Specifically, we employed *ground truth human point clouds* as the 3D Gassian positions in order to eliminate any confounding impacts that might arise from an inaccurate 3D Gaussian position. The two experimental settings are: **1**). Training a generalizable model that predicted all 3D

Table 1: The results of different settings on Thuman. Results: Ground truth point clouds are used. *Results: generated point clouds are used.*

| | | | PSNR | SSIM | LPIPS |
|---|---|---|---|---|---|
| indirect | | | 31.84 (29.02) | 0.963 (0.953) | 0.056 (0.070) |
| direct | independent | vanilla | FAIL | FAIL | FAIL |
| | | neural | 32.99 (29.53) | 0.968 (0.949) | 0.052 (0.069) |
| | joint | | 33.22 (29.71) | 0.971 (0.951) | 0.048 (0.065) |

Gaussian attributes using the splat-based rasterizer for supervision (**indirect** in Tab. 1), **2**). Training a generalizable model that predicted all 3D Gaussian attributes, utilizing the proxy ground truth 3D Gaussian attributes for L1 loss supervision. (**joint** in Table 1; note that we trained a regression model instead of a diffusion model due to the limitations of the *indirect paradigm* for a fair comparison). As illustrated in Tab. 1, relying solely on differentiable rendering to supervise the training process results in the poorest performance. Furthermore, as presented in Fig. 4(a), the result of the indirect paradigm was blurry and exhibited sharp, incorrect colors. We hypothesized that this was due to

---

[1]We found that the voxel size of the point transformer will seriously affect the performance. Hence, we did not use it as the backbone of HUMAN-DAD.

Table 2: Quantitative comparisons of different methods on Thuman, CityuHuman, 2K2K, and CustomHuman datasets. The best results are highlighted in **bold**.

| Metric / Method | Thuman | | | CityuHuman | | | 2K2K | | | CustomHuman | | |
|---|---|---|---|---|---|---|---|---|---|---|---|---|
| | PSNR | SSIM | LPIPS | PSNR | SSIM | LPIPS | PSNR | SSIM | LPIPS | PSNR | SSIM | LPIPS |
| GTA (NeurIPS 2023) | 25.78 | 0.919 | 0.085 | 27.41 | 0.923 | 0.075 | 24.15 | 0.921 | 0.080 | 28.86 | 0.920 | 0.088 |
| SiTH (CVPR 2024) | 25.36 | 0.919 | 0.083 | 29.21 | 0.934 | 0.067 | 24.30 | 0.920 | 0.076 | 26.47 | 0.911 | 0.095 |
| LGM (ECCV 2024) | 25.13 | 0.915 | 0.096 | 29.78 | 0.941 | 0.074 | 27.99 | 0.938 | 0.071 | 31.91 | 0.944 | 0.077 |
| SHERF (ICCV 2023) | 26.57 | 0.927 | 0.081 | 30.13 | 0.942 | 0.067 | 27.29 | 0.931 | 0.072 | 27.88 | 0.916 | 0.096 |
| HUMAN-DAD | **30.03** | **0.953** | **0.065** | **32.47** | **0.954** | **0.062** | **30.64** | **0.949** | **0.060** | **34.82** | **0.958** | **0.055** |

the rendered images being generated by the strongly coupled attributes of 3DGS; only image loss could not provide accurate attribute-wise gradients, which made the network lean to predict smooth attributes. This hypothesis is supported by the results of Setting 2. Under the supervision of the proxy ground truth 3D Gaussian attributes, we could decouple different attributes and provide an attribute-wise gradient for backpropagation. The results show that using our pseudo dataset for supervision is more effective than differentiable rendering, and the construction of the proxy ground truth 3D Gaussian attributes is necessary.

**Defect of Vanilla-3DGS.** The solution space of vanilla-3DGS is expansive, primarily due to its direct optimization of numerical values and the strong coupling among its various attributes. To validate this, we designed three experiments. Firstly, we fixed all random seeds of python, numpy and pytorch and optimized the 3DGS model on a specific scene 200 times. Subsequently, we plotted the results of the 200 optimizations, which were obtained by summing the spherical harmonic values from each optimization, as depicted in Fig. 4(b). The results exhibit significant variation. Secondly, we selected a local area (marked in yellow) on a human body expected to have the same color; however, as illustrated in Fig. 4(c), the spherical harmonic values ranged widely from -12.5 to 6. Lastly, we visualized the spherical harmonic, opacity, and scale attributes in Fig. 4(d) left column, revealing that vanilla-3DGS produced highly chaotic results. Although vanilla-3DGS can achieve better rendering results, it is impractical to learn from something that is filled with randomness and lacks regularity (as illustrated in Tab. 1). Therefore, it is not suitable for constructing the proxy ground truth 3D Gaussian attributes.

**The Necessity of two-stage construction.** We visualized the results of the per-scene overfitting stage in Fig. 4 (c) and (d) right column. Our point transformer-based 3DGS significantly narrowed the variation range of spherical harmonic values in the local area, and the visualizations of the spherical harmonic, opacity, and scale attributes appeared cleaner and more uniform. However, as depicted in Fig. 5, the obtained 3D Gaussian attributes from the first stage lead to a slow convergence during training, primarily due to the independent optimization of scenes, which results in distinct distributions across different scenes. Moreover, the distribution among various scenes had been further aligned after the second stage, with the variances of the minimum and maximum values reduced from **0.0707** to **0.0329** and from **0.1114** to **0.1094**, respectively. As shown in Tab. 1, the two-stage construction (**joint**) achieves better performance than the single-stage construction (**neural**). Hence, it is necessary to conduct a two-stage construction process.

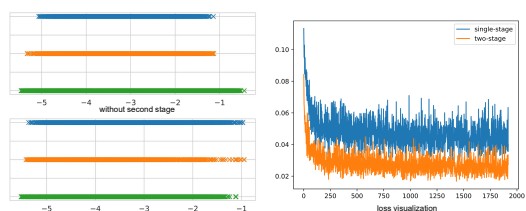

Figure 5: The visualization of distributions and loss. 🔍 Zoom in for details.

### 4.3 COMPARISONS WITH STATE-OF-THE-ART METHODS

We compared our HUMAN-DAD with four state-of-the-art methods: GTA (Zhang et al., 2023), LGM (Tang et al., 2024), SiTH (Ho et al., 2024), and SHERF (Hu et al., 2023). The four used datasets were collected from individuals of different ages, genders, and races. As reported in Tab. 2, HUMAN-DAD achieves the best quantitative performance across all metrics on all datasets, underscoring the effectiveness and generalization capability of HUMAN-DAD.

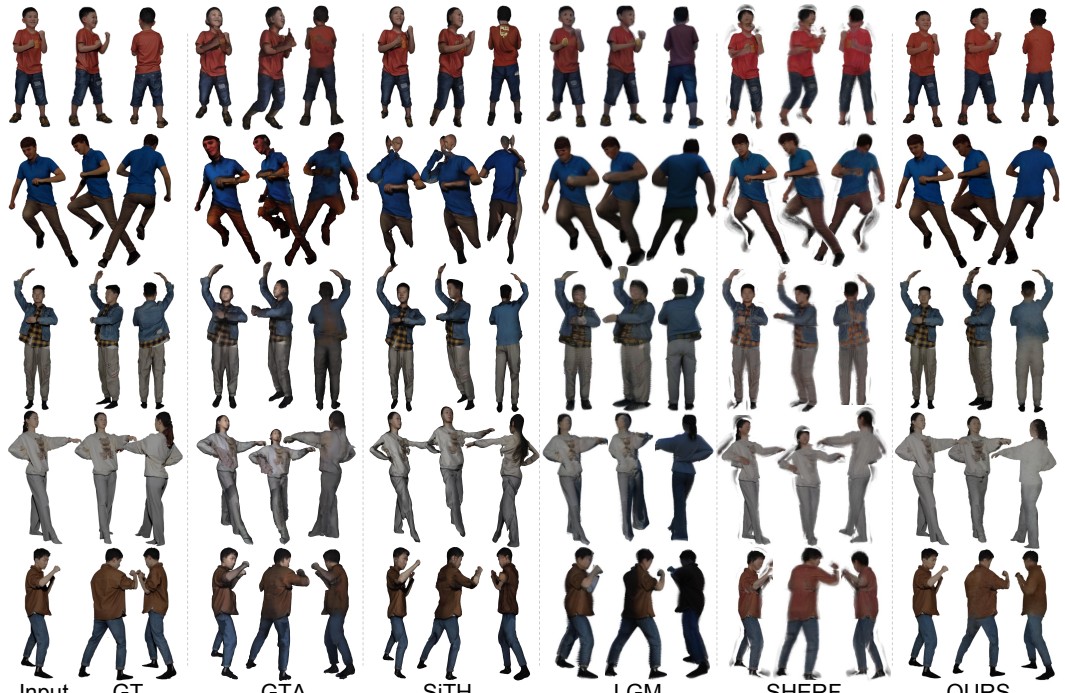

Figure 6: visual comparisons of our method with GTA, SiTH, LGM, and Sherf. 🔍 Zoom in for details.

GTA and SiTH suffer from the grid resolution of the marching cube and produce broken reconstructed human bodies, resulting in low-fidelity novel views. Moreover, their results are usually in the wrong poses. However, as shown in Fig. 6, our HUMAN-DAD is capable of rendering fine-grained input view images while maintaining texture consistency across different view directions with correct poses.

LGM is a generalizable 3DGS model that predicts the 3D Gaussian attributes from the multi-view generated images. However, it occasionally predicts incorrect backside images, and the rendered images have low resolution due to consistency issues across different views. Our HUMAN-DAD solves the consistency problem by generating the 3D Gaussian position first. Moreover, owing to our diffusion-based framework, the rendered images are more realistic, compared to LGM.

SHERF frequently encounters challenges with incorrect poses in SMPL models. While the estimated SMPL models often exhibit poses similar to the given view on the 2D plane, their poses are often incorrectly along the z-axis, and SHERF lacks a module to rectify the SMPL models in 3D space. Additionally, SHERF heavily relies on the SMPL model and is unable to render loose clothing, as illustrated in Figure 6, where it fails to render the hems of the jeans for the first individual. Furthermore, SHERF directly utilizes the input view to extract image features, which leads to incorrect backside information prediction when the backside of the human is not symmetrical with the frontal view. Our designed 3D Gaussian position generator and pixel-aligned feature can tackle these issues.

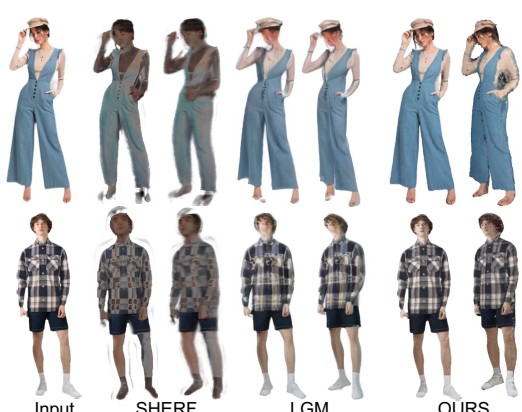

Figure 7: The visual comparison with SHERF and LGM on the wild images. 🔍 Zoom in for details.

We also compared HUMAN-DAD with SHERF and LGM on the wild images. As shown in Fig. 7, SHERF cannot correctly recover the colors, and the results were in the wrong poses. Moreover, it

Figure 8: The visual results of ablation studies. (a) regression-based model (both direct and indirect paradigms) v.s. diffusion-based model. (b) without backside image v.s. with backside image. (c) without SMPL semantic features v.s. with SMPL semantic features. 🔍 Zoom in for details.

cannot recover loose clothing. LGM is able to recover the correct colors; however, it cannot preserve the face identity and the texture details. We also refer reviewers to the *Supplementary Material* for the video demo and *Appendix* Fig. 14 for more visual results.

## 4.4 ABLATION STUDIES

**Diffusion-based Model.** Thanks to the novel direct paradigm we proposed, simply training a regression-based model also exhibits good quantitative performance on the Thuman dataset, as demonstrated in Tab. 3. However, the regression models (both direct and indirect paradigms) usually overfit in local-optima and struggle to predict accurate texture in occluded areas, as illustrated in Fig. 8(a). Conversely, this issue can be addressed by the diffusion-based model. Hence, we consider the direct paradigm to be better because it supports training diffusion models, which are more efficient for learning the distribution of the 3D Gaussian attributes.

**Backside Image.** The predicted backside image provides coarse information about unseen areas for HUMAN-DAD. When the backside image is deprecated, as illustrated in Fig. 8(b), we observed that HUMAN-DAD failed to predict the correct face identity, and the quantitative performance also decreased. These results demonstrate that the backside image actually contributes to improving the quality of unseen areas.

**SMPL Semantic Feature.** As presented in Fig. 8(c), when the SMPL semantic feature is not adopted, we observed that the boundary of the neck area became unclear, and the left hand appeared darker compared to the full model. This over-smoothing effect impacts the quantitative performance. We believe that the SMPL-semantic feature enables HUMAN-DAD to learn the human body structure and tackle the over-smoothing problem.

Table 3: The results of ablation studies on Thuman.

| | | Back Image | Smpl Semantic | PSNR | SSIM | LPIPS |
|---|---|---|---|---|---|---|
| indirect | | ✔ | ✔ | 29.02 | 0.953 | 0.070 |
| direct | regression | ✔ | ✔ | 29.71 | 0.951 | 0.065 |
| | diffusion | ✗ | ✗ | 28.44 | 0.948 | 0.074 |
| | | ✔ | ✗ | 29.63 | 0.950 | 0.070 |
| | | ✔ | ✔ | 30.03 | 0.953 | 0.065 |

## 5 CONCLUSION

We have introduced a novel *direct paradigm* for training a generalizable 3DGS model of the human body. In crafting this *direct paradigm*, we implemented a two-stage process to acquire proxy ground truth 3D Gaussian attributes. Furthermore, we devise a diffusion model to capture the global distribution of 3D Gaussian. Extensive experimental results have demonstrated the significant superiority of HUMAN-DAD over the current state-of-the-art methods. We will explore the potential of HUMAN-DAD to support multi-view images (Kwon et al., 2021; Gao et al., 2023). At present, the direct paradigm has only been applied to the monocular human rendering task. In future work, we intend to explore and validate its effectiveness in a broader range of 3DGS application scenarios.

ETHICS STATEMENT

We have read and adhere to the Code of Ethics of ICLR 2025. This study does not encounter violations of the Code of Ethics.

REPRODUCIBILITY STATEMENT

we provide detailed information on the implementation and network architectures. We also thoroughly explain the datasets and evaluation metrics used in our study. To ensure accuracy and reproducibility, we conduct extensive experiments on the Thuman2 (Yu et al., 2021), CityuHuman (Tang et al., 2023), 2K2K (Han et al., 2023), and CustomHuman (Ho et al., 2023) datasets. Additionally, if our paper is accepted by ICLR 2025, we will release the source code on GitHub.

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

# A APPENDIX

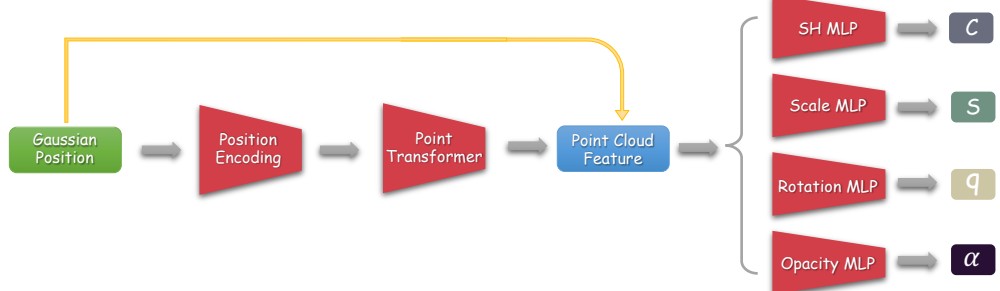

Figure 9: The architecture of point transformer $\Theta_1$.

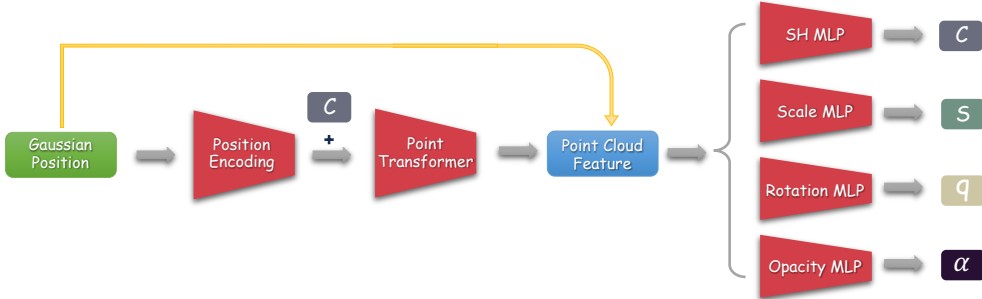

Figure 10: The architecture of point transformer $\Theta_2$.

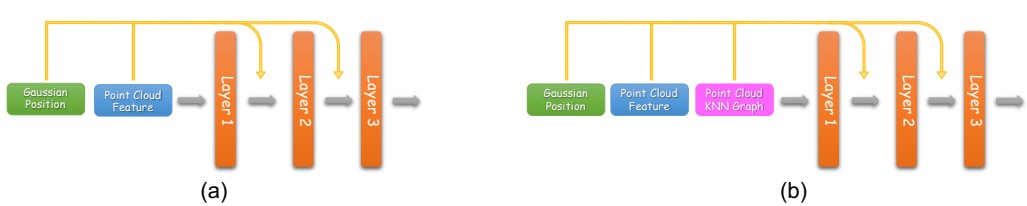

Figure 11: The architecture of MLPs. (a). The architecture of Spherical Harmonics MLP. (b). The architecture of Scale, Rotation and Opacity MLPs.

## A.1 ARCHITECTURE OF POINT TRANSFORMER

We present the architectures of the point transformers, denoted as $\Theta_1$ and $\Theta_2$, in Fig. 9 and Fig. 10, respectively. The architecture of the MLPs is depicted in Fig. 11. Initially, the human point cloud is fed into a position encoding module to enable the network to learn high-frequency features. Subsequently, a point transformer is employed to extract point-wise features from the human point cloud. These point-wise features are then concatenated with the human point cloud and input into various MLPs to learn different 3D Gaussian attributes. In the second stage of overfitting, we also incorporate spherical harmonics features into the point transformer to unify the distribution across different scenes. For the MLPs responsible for Scale, Rotation, and Opacity, we further enhance their geometric perception by inputting the KNN graph of the point cloud.

## A.2 ARCHITECTURE OF 3DGS DIFFUSER

We present the process of preparing the condition features for the 3DGS diffuser in Fig. 12. Currently, most point cloud-based diffusion models are designed for point cloud generation, yet they

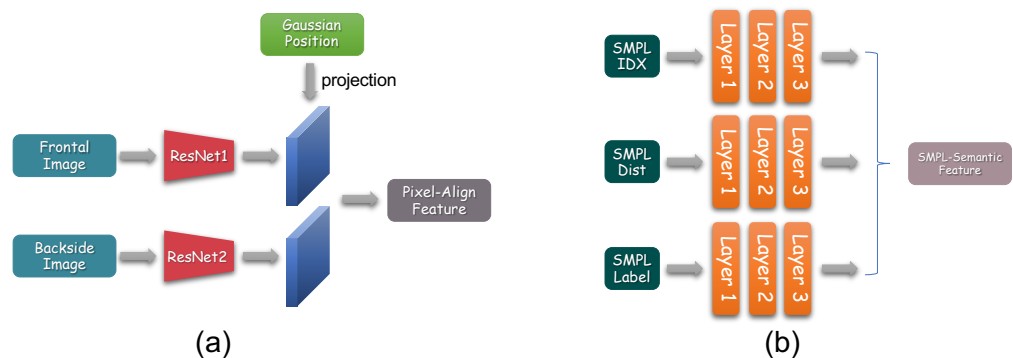

Figure 12: The process of preparing the pixel-aligned feature and the smpl-semantic feature. (a). The process of preparing the pixel-aligned feature. (b). The process of preparing the smpl-semantic feature.

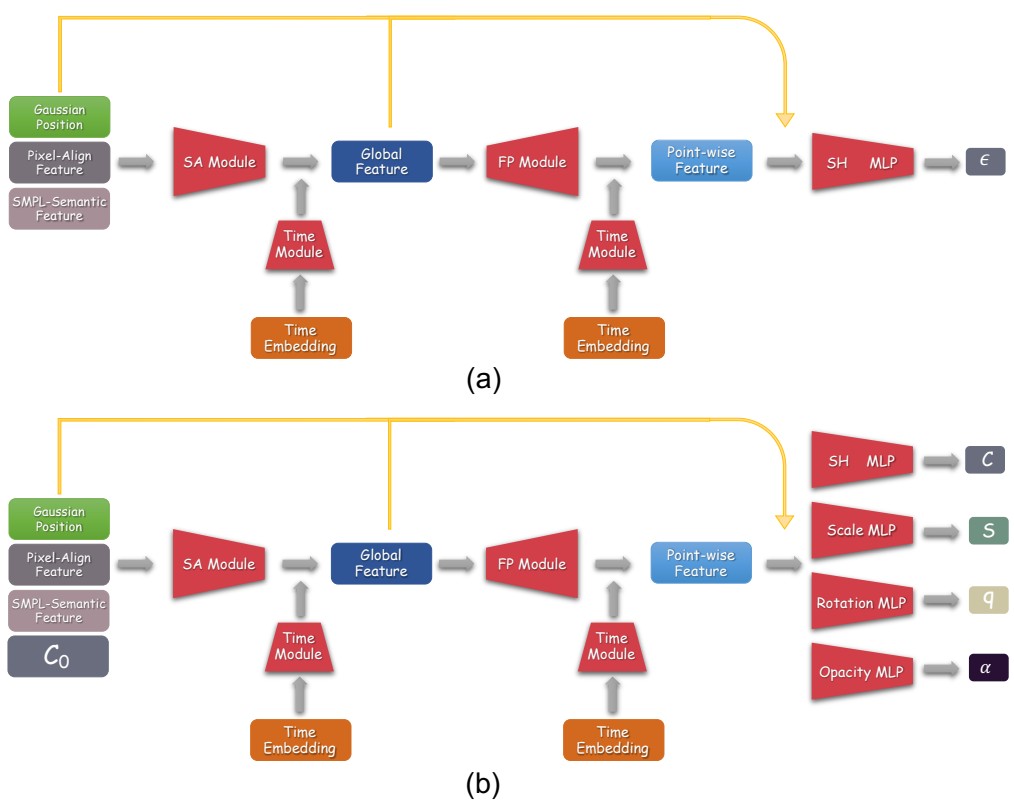

Figure 13: (a). The architecture of the 3D Gaussian attribute diffusion model. (b). The architecture of the 3D Gaussian attribute diffusion model to train the extra step.

lack the capability to directly apply diffusion on 3D Gaussian attributes. To address this, we adopt PointNet++ as the backbone and introduce modifications to enable the training of a diffusion model. The architecture of the diffuser is illustrated in Fig. 13.

### A.3 SUBJECTIVE EVALUATION

We conducted a subjective evaluation to compare various methods quantitatively. Specifically, we engaged 52 participants, including undergraduate students, postgraduate students with diverse research backgrounds, and professionals from the industry, to assess 8 different human bodies. For each human body, we presented three images generated by different methods and requested the par-

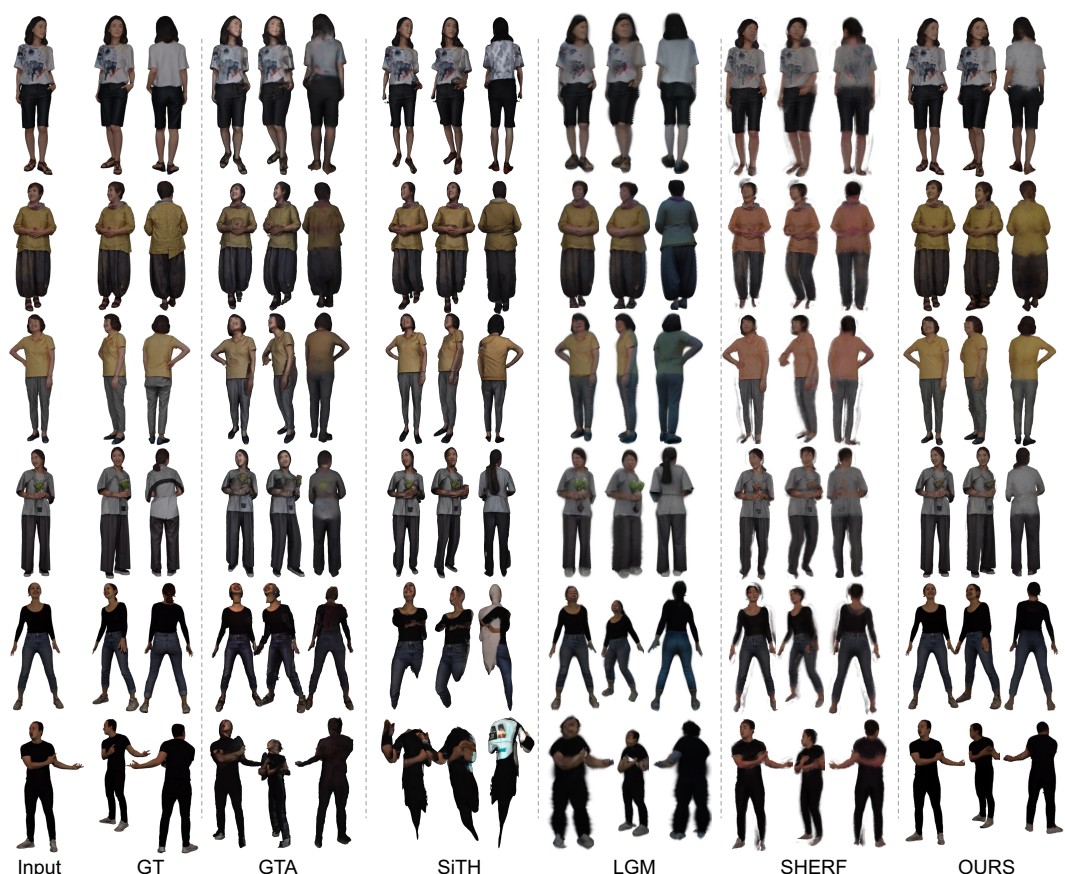

Input      GT      GTA      SiTH      LGM      SHERF      OURS

Figure 14: More visual results.

ticipants to provide scores within the range of 1 to 5, reflecting the quality of the generated shapes, with ratings as follows: 1: poor, 2: below average, 3: average, 4: good, 5: excellent. Fig. 15 shows the results of the subjective evaluation, where we provided the overall scores, the mean value, and the standard deviation (std) of the scores. It can be seen that our HUMAN-DAD obtains the highest mean score.

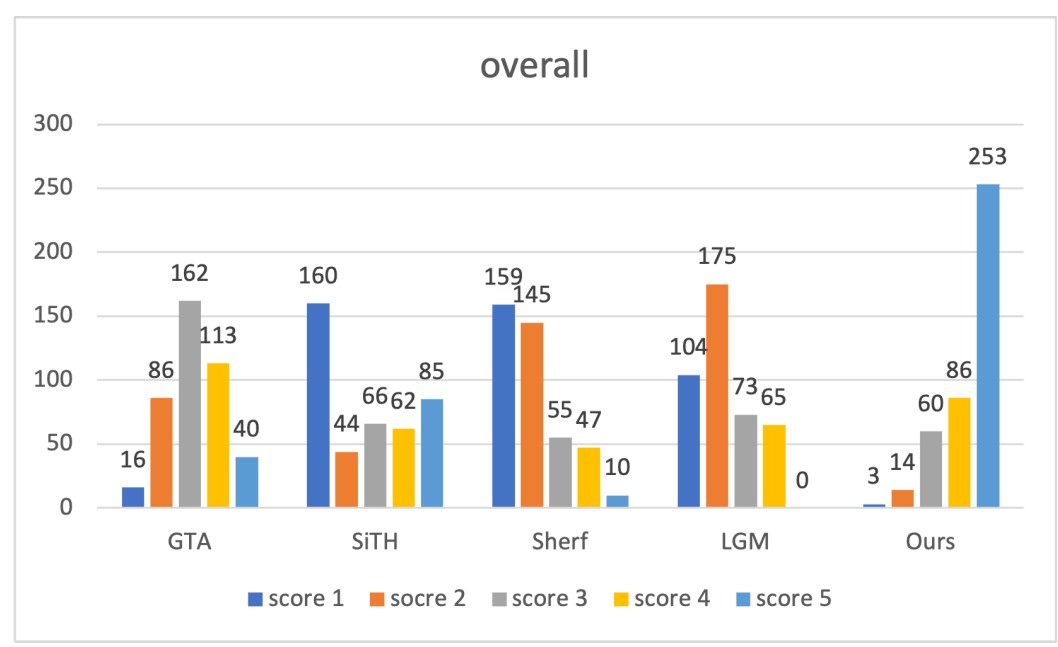

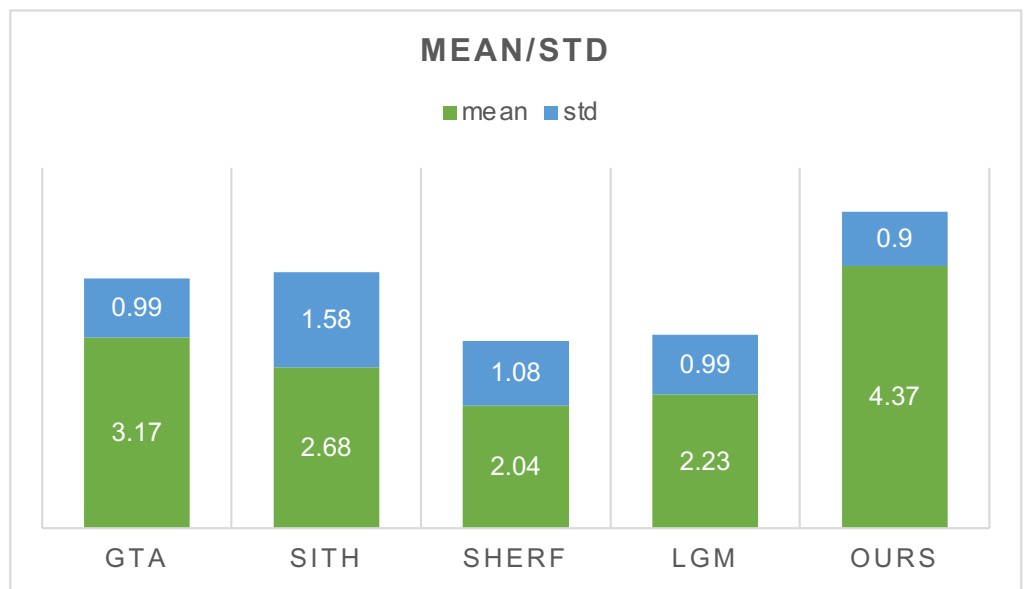

Figure 15: Overall and mean/std results of the subjective evaluation.

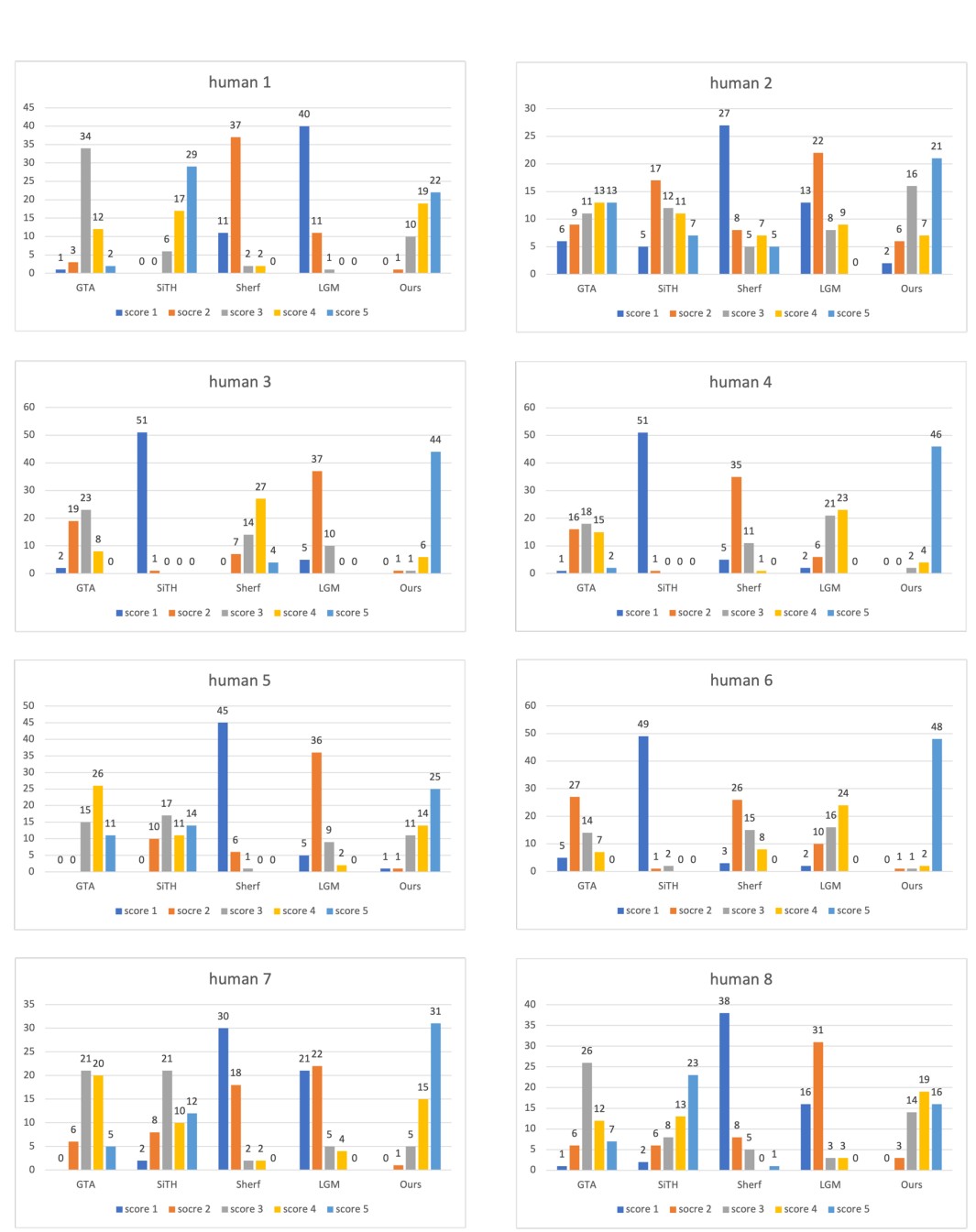

Figure 16: Results of the subjective evaluation of each human.

REBUTTAL CONTENT

**Results of Animation.** We present several results of novel pose synthesis (animation) in Fig. 17. By blending the 3D Gaussian positions with the SMPL vertices and modifying the SMPL pose, we successfully achieve novel poses. Notably, the results are satisfactory despite the absence of a specifically trained model for this task.

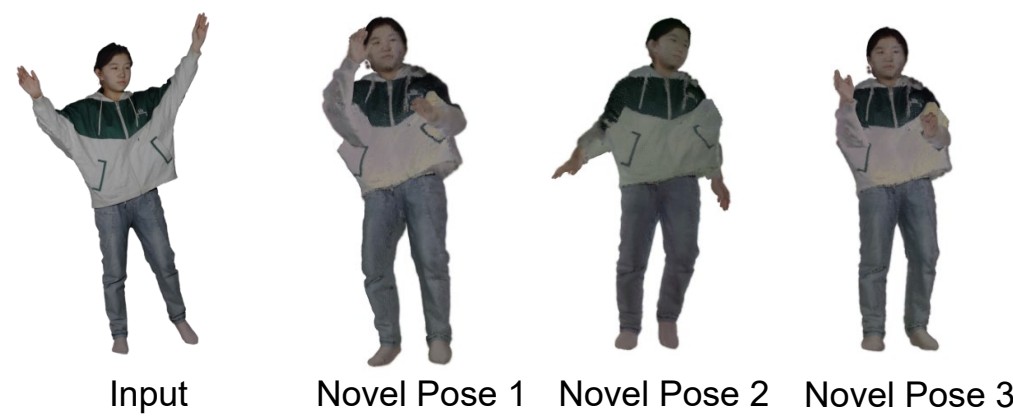

| Input | Novel Pose 1 | Novel Pose 2 | Novel Pose 3 |

Figure 17: Novel pose synthesis results of HUMAN-DAD. Zoom in for details.

**Results of Image-conditioned Diffusion model.** We explored the use of a diffusion model conditioned solely on the input image to predict 3D Gaussian positions, SHs, rotations, scales, and opacities. However, as shown in Fig. 18, we found that such a model failed to train effectively, indicating that using only an image as a condition is insufficient for training a robust diffusion model. This highlights the importance of designing HUMAN-DAD.

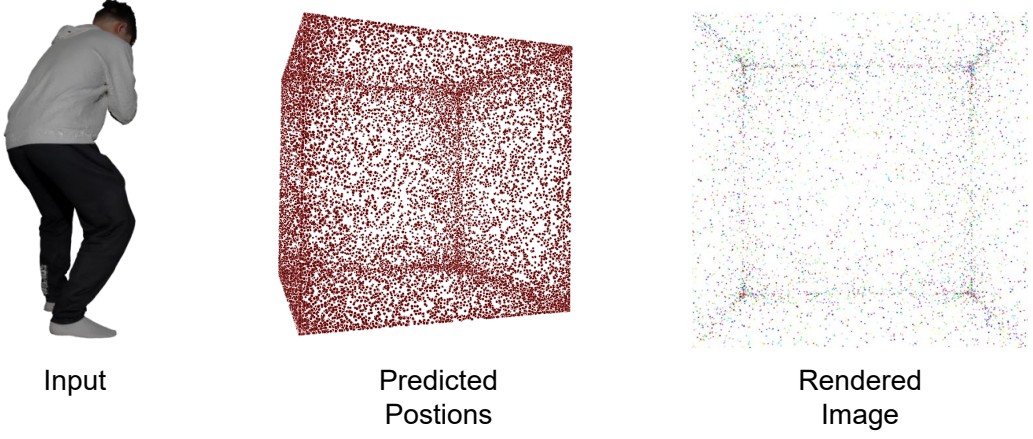

| Input | Predicted Postions | Rendered Image |

Figure 18: The results of the image-conditioned diffusion model.

**Appling the two-stage construction process on other datasets.** We present two examples of fitting results on the 2K2K and CustomHuman datasets in Fig. 19. These datasets, unlike Thuman, were collected from different countries and feature individuals of varying ages and races. Despite these differences, our two-stage construction method proves to be effective.

**Vanilla-3DGS in the first stage.** As shown in Fig. 20, using the vanilla-3DGS in the first stage results in a final output that fails to preserve fine details. This is due to the high level of randomness in the vanilla-3DGS, which the neural network in the second stage is insufficiently robust to fully eliminate. However, using a neural network in the first stage will tackle the randomness problem.

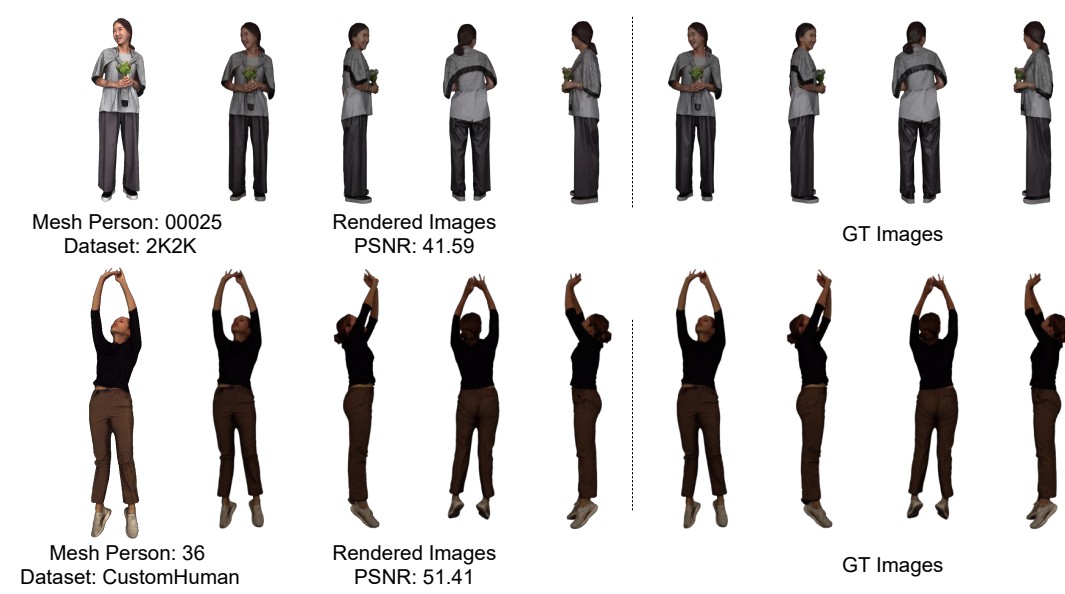

Figure 19: The fitting results of the two-stage construction process on 2K2K and CustomHuman datasets.

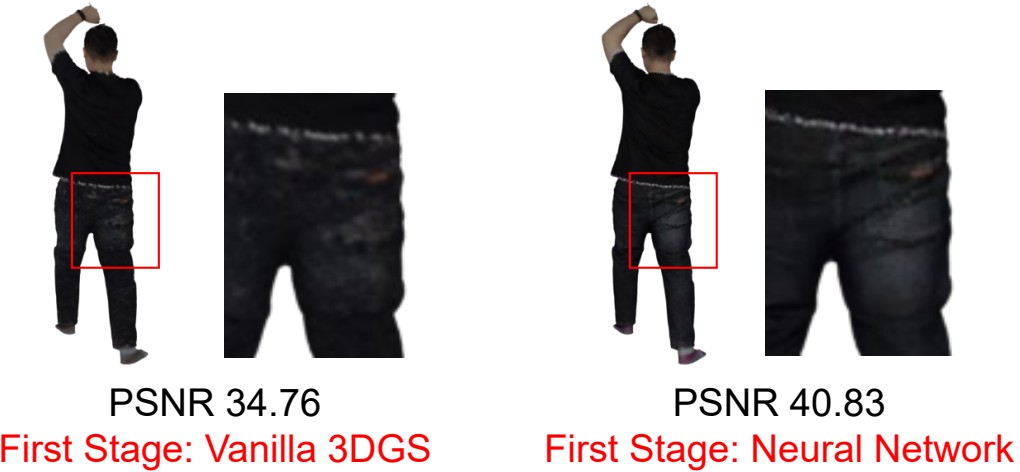

Figure 20: Visual results of using vanilla-3DGS and neural network at the first stage, respectively. Zoom in for details.

**Evaluation with ground truth 3D shapes.** We provided ground truth occupancy fields, SDF fields, and point clouds for GTA, SiTH, and HUMAN-DAD, respectively. (LGM learns 3D Gaussian positions from predicted multi-view images, while SHERF requires SMPL.) As shown in Fig. 21, both GTA and SiTH fail to capture fine details, whereas HUMAN-DAD demonstrates superior performance in preserving intricate features.

**Optimizing 3DGS with only two images.** We optimized a 3DGS model with only the frontal and back images, as shown in Fig. 22, the 3DGS model is too random, while our HUMAN-DAD can reduce the randomness problem.

**Comparison between Human3Diffusion and Human-DAD.** We compared Human3Diffusion with HUMAN-DAD, as illustrated in Fig. 23. The results show that our HUMAN-DAD effectively preserves fine details, whereas Human3Diffusion falls short in this regard.

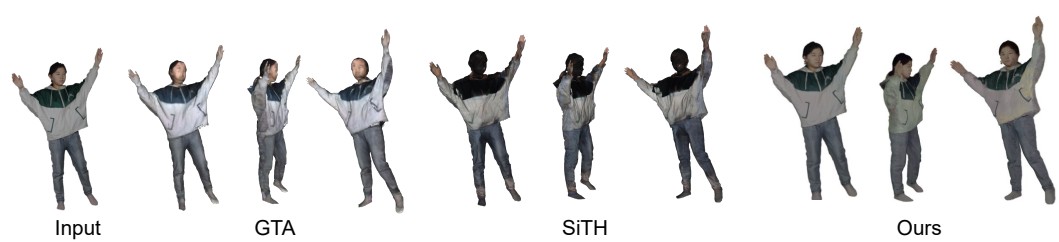

Input      GTA      SiTH      Ours

Figure 21: Visual comparison between GTA, SiTH and Human-DAD when using ground truth 3D shapes. Zoom in for details.

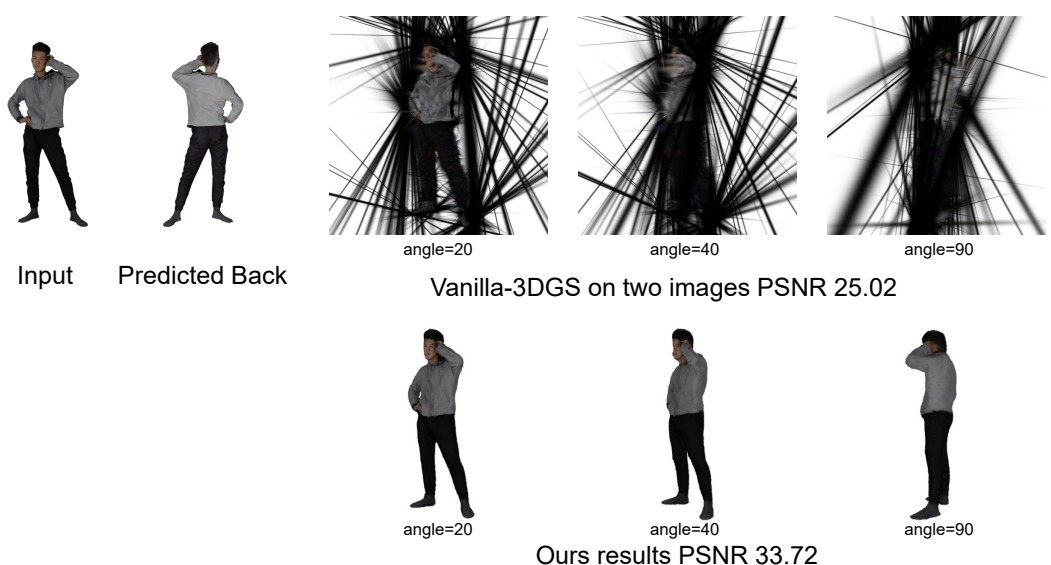

Input     Predicted Back

Vanilla-3DGS on two images PSNR 25.02

Ours results PSNR 33.72

Figure 22: Visual results of only using frontal and back images to optimize a 3DGS model.

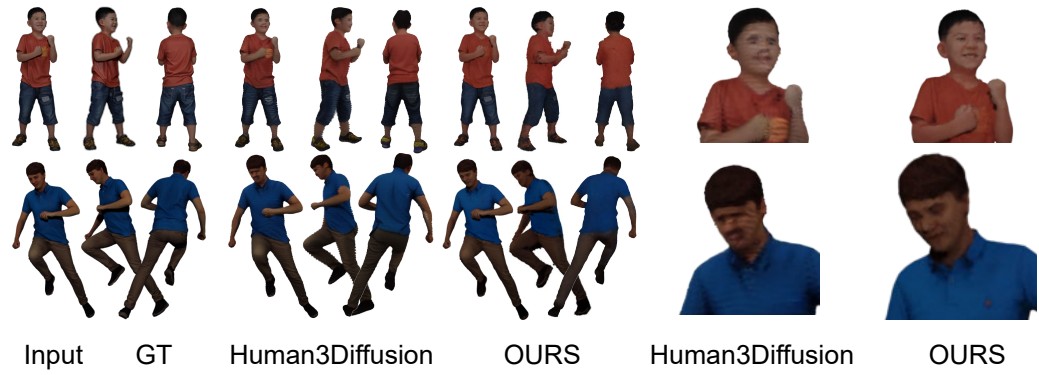

Input    GT    Human3Diffusion    OURS    Human3Diffusion    OURS

Figure 23: Visual comparison between Human3Diffusion and Human-DAD. Zoom in for details.

