# OpenReview forum: "Generalizable Monocular 3D Human Rendering via Direct Gaussian Attribute Diffusion"
_ICLR.cc/2025/Conference — Submitted to ICLR 2025_

### Official Review · Reviewer_73DC · 2024-10-28

**Soundness:** 2
**Presentation:** 3
**Contribution:** 2
**Rating:** 3
**Confidence:** 5

**Summary:**

This paper proposes Human-DAD, which adapts a new paradigm for training a generalizable 3DGS, to reconstruct the 3D human body from a single-view image. Unlike a common strategy in optimizing 3DGS from multi-view images, the authors proposed to train the model with direct supervision (applying loss function directly into Gaussian attributes). To apply good supervision, this paper uses two-stage workflow to construct 3D pseudo-GTs of 3D Gaussians: optimizing 3D Gaussian per scene and then unifying the distribution of Gaussian attributes along the camera views. Furthermore, to reconstruct 3D human Gaussians from the single-view image, the authors device a diffusion model conditioned from SMPL-semantic features and pixel-aligned features. The proposed Human-DAD outperforms existing state-of-the-art 3D human reconstruction methods.

**Strengths:**

In recent years, 3D Gaussian splatting (3DGS) has shown impressive results in 3D vision fields. Despite the promising results of 3DGS, the exploration of adapting 3DGS for single-view 3D human reconstruction is still lacking. This paper can provide several good intuitions about the 3DGS for 3D human reconstruction, especially in the proposed 'distribution unification' of 3D Gaussian pseudo-GTs.

**Weaknesses:**

I have several concern points as below.


1) Discussion about direct vs indirect supervision

a. This paper strongly suggests the inefficacy of the traditional 'indirect paradigm' in reconstructing 3D humans with 3DGS.
In 3DGS for 3D objects, I agree that recent works apply image loss to train their model instead of direct supervision [4]. However, to the best of my knowledge, I haven't heard about employing 3DGS for single-view 3D human reconstruction (also in the submitted paper). If there is a reference algorithm, it would be good to refer to it and conduct the experiment based on it. If not, the proposals in this paper may be a hasty conclusion.

b. This paper validates the effectiveness of the direct paradigm in their constructed algorithm (Tab. 1). However, their experiment setting in Tab. 1 cannot be representative of covering recent 3D human reconstruction approaches. So, it is difficult to suggest that a direct paradigm is better by only referring Tab. 1. I recommend that experiment Tab. 1 should be conducted in more general settings, such as [1].

c. Direct supervision is not always better than indirect supervision. In the two-stage workflow (Fig. 2), the algorithm reduces the randomness of fitted 3DGS, but it does not completely solve it. In generating 3D pseudo-GTs, enormous 3DGS can be mapped into multi-view images, similar to many-to-one mapping (L44) in indirect supervision. This discussion needs to be taken into consideration


2) Distribution unification

As shown in Fig. 4 and Tab. 1, the distribution unification (L250-257) is effective. However, it is difficult to understand why the training approach of Fig. 2 (right) helps the Gaussian distribution. The approach is very simple, batch-wise training. For example, if the network learns identity (A -> A), it would be a good local minima to minimize the photometric loss. What is the main component that to leads unifying the Gaussian distribution? A detailed discussion of this should be included in the manuscript.


3) Evaluation of 3D geometry

This paper only produces evaluation in rendered images (PSNR, SSIM, ...) and does not report 3D metrics (P2S, Chamfer Distance, ...), although the used datasets have 3D GT scans.


4) Paper writing

a. In L32-38, it does not clarify the categorization: Reconstruction-based vs NerF-based. The reconstruction-based approaches also use neural fields. I think that more suitable words are image-based vs video/multi-view-based. I recommend that this paper needs more clarification about it.

b. In L128, the paper mentioned Zou et al. follow PIFu. It's not clear which part of PIFu they followed (PIFu does not use triplane representation).

c. It is unclear what the authors used datasets for Tab. 1 and Tab. 3.


[1] Zou et al., Triplane meets gaussian splatting: Fast and generalizable single-view 3d reconstruction with transformers, CVPR, 2024.


In conclusion, I'm concerned about accepting this paper, for two main reasons.
a. It is hasty to suggest direct supervision is beneficial, given the content of this paper.
b. Distribution unification of 3DGS is difficult to agree on, without extensive explanation of its design reason.

**Questions:**

1) Diffusion model
It is unclear why the diffusion model was chosen. I conjecture the main strength of the diffusion model is generative power.
Does the algorithm produce diverse reconstruction candidates corresponding to the single-view image?

2) Reconstruction quality
In Fig. 8, why the reconstructed human face is blurred, unlike Fig. 6?
In Fig. 7, why the SHERF's results are more darker than others, unlike Fig. 6?

---

> ### Author Response · Authors · 2024-11-23
>
> ### **[Rebuttal to Reviewer 73DC]**
>
> ### **[W1]** *Discussion about direct vs indirect supervision.*
> **Response:**
>
> a.We kindly **disagree** with your conclusion that the direct paradigm is a hasty conclusion. As you mentioned, there are no methods using 3DGS for single-view human reconstruction; hence, we build an indirect paradigm baseline to validate the effectiveness of our proposed direct paradigm. Note that, even the indirect paradigm achieves the best performance when compared with other methods. Both quantitative and qualitative results strongly support that our direct paradigm is better than the indirect paradigm.
>
> Moreover, the indirect paradigm typically only supports regression training, however, **such a direct paradigm supports training a diffusion model which is more efficient than the regression-based model to learn the 3D Guassian attribute distribution (see Figure 8).**
>
> b. We have quantitatively compared the results of indirect and direct paradigms in Table 1 and Table 3.
> Our proposed direct paradigm achieves better performance than the indirect paradigm, moreover, the visual
> results in Figure 4(a) and Figure 8(a) shows that the direct paradigm can achieve better visual quality. These results support that
> our direct paradigm is better than the indirect paradigm.
>
> c.The randomness of the vanilla-3DGS is very large, the two-stage workflow aims to reduce the randomness of the 3DGS as much as possible, we never claim that we can completely solve it. When the randomness of the indirect paradigm is reduced, we can learn the model easier in a smaller solve space with the direct paradigm. Note that, **there won’t be enormous points** because the total number of 3DGS is **fixed as N** (according to Eq. 7).
> ### **[W2]** *Distribution unification.*
> **Response:** Thank you for your feedback. I believe there is a **significant misunderstanding** regarding our paper, particularly with the **interpretation of Figure 4**. To clarify, Figure 4 does not illustrate the effectiveness of the distribution unification stage (L400-401 in the original paper). Instead, the left column of Figure 4(d) shows the result of the vanilla-3DGS, while the right column shows the result after the per-scene overfitting stage (the first stage).
>
> Our proxy-3DGS ground truth construction consists of two stages: the per-scene overfitting stage and the distribution unification stage.
>
> Per-scene overfitting stage: In this stage, the vanilla 3DGS results are highly random and the internal distribution of each individual 3DGS is non-uniform (left column of Figure 4(d)). To address this, we leverage the smoothness of the neural network (as noted by reviewer jqqm) to regularize the optimization process. This allows us to achieve a more uniform and structured 3DGS, which is shown in the right column of Fig. 4(d).
>
> Distribution unification stage (Figure 5): Although the per-scene overfitting stage improves the internal structure of individual 3DGSs, the external distribution among different 3DGSs remains non-uniform, as each 3DGS is optimized independently. To further align the distribution across different 3DGSs, we introduce the distribution unification stage. This second stage also utilizes the smoothness of the neural network to ensure that the 3DGSs from different scenes share a more consistent and unified distribution. **More details and explanations about the distribution unification stage can be found in Figure 5 and L400-416 in the original paper**.
>
> ### **[W3]** *Evaluation of 3D geometry.*
> **Response:** As you suggested, we have conducted the geometry evaluation on the Thuman and CustomHuman datasets. (SHERF does not reconstruct the 3D geometry). As reported in the Table below, our Human-DAD can have much better 3D geometry reconstruction performance than other methods. Even so, we want to emphasizethat this work focuses on the novel view synthesis, the reconstructed 3D geometry is not the objective of this paper.
> | **Method \ Metric** | **Chamfer Distance (Thuman)** | **P2S (Thuman)** | **Chamfer Distance (2K2K)** | **P2S (2K2K)** |
> |----------------------|------------------------------|------------------|----------------------------|----------------|
> | GTA (NeurIPS 2023)  | 1.040                        | 1.020            | 1.195                      | 1.158          |
> | SiTH (CVPR 2024)    | 0.879                        | 0.932            | 1.076                      | 1.137          |
> | LGM (ECCV 2024)     | 1.412                        | 1.163            | 1.007                      | 0.864          |
> | **Human-DAD**       | **0.451**                    | **0.471**        | **0.758**                  | **0.759**      |

---

> > ### Comment · Reviewer_73DC · 2024-11-23
> >
> > Thank you for your detailed and kind answer.  I feel like some of my questions were misrepresented, so I'll kindly ask them again.
> >
> > ## [W1] Discussion about direct vs indirect supervision.
> >
> > My main concern is the hasty generalization about direct vs indirect paradigm.
> >
> > I agree that there is no way to compare other 3DGS + single view human reconstruction methods, and I don't doubt your direct vs indirect ablation study.
> >
> > My concern is that the paper is drawing conclusions about the direct vs. indirect paradigm based on relatively few experiments.
> >
> > The direct paradigm is beneficial for the pipeline you designed, but may not be beneficial for other approaches.
> >
> > Your designed pipeline is not representative method of 3DGS.
> >
> > The best thing to do is to experiment with the direct vs indirect paradigm for all the settings that 3DGS is capable of.
> > However, this paper does not adequately cover this experiment.
> >
> > I think the direct vs. indirect paradigm is still a matter of debate in the 3D human reconstruction community.
> >
> > (I think reviewer GRmh also felt unclear at this point.)
> >
> > Accordingly, I agree that your proposed method achieves good performance, but the claims in the paper are still unacceptable and unclear.
> >
> >
> > ## [W2] Distribution unification.
> >
> > My main question is why distribution unification works so well.
> >
> > In your rebuttal, you suggest that 'smoothness of the neural network' is helpful for the distribution unification.
> >
> > What is 'smoothness of the neural network'?
> >
> > I'm still confused about how distribution unification works.
> >
> > As I understand it, distribution unification means that the attributes of 3D Gaussian do not change even if the rendering view changes.
> >
> > For example, the first Gaussian of A_1 indicates left shoulder and the first Gaussian of A_2 also indicates left shoulder.
> >
> > How do you guarantee this?
> >
> > Even though you simply do end-to-end training. (Eq. 9) without any constraint, it is surprising and puzzling that the distribution is consistent.
> >
> > I need a clearer explanation of it.

---

> > > ### Author Response · Authors · 2024-11-24
> > >
> > > # **Discussion about direct vs indirect supervision.**
> > >
> > >
> > > We provided a toy experiment to further explain why the direct paradigm is more efficient.
> > >
> > > The objective of a simple neural network in this experiment is to produce an **output** tensor with the shape (256, 1), such that the sum of its elements equals 256 (**output.sum() = 256**).
> > >
> > > ```python
> > > input = torch.randn(256)
> > > output = toymodel(input)
> > > ```
> > >
> > > There are numerous solutions to achieve a sum of 256. For example:
> > >
> > > - **Solution 1**: All elements are equal, e.g., `[1, 1, 1, ..., 1]` (256 elements of value 1).
> > > - **Solution 2**: Non-uniform distribution, e.g., `[2, 2, 2, ..., 0.5, 0.5]` (most elements are 2, with a few elements summing to the remainder).
> > > - **Solution 3**: Random distribution, e.g., `[10, 5, 0, ..., 0.1, 1.9]` (values vary widely but still sum to 256).
> > >
> > > This highlights the **many-to-one problem**, as multiple configurations of the tensor can achieve the same target sum, making supervision ambiguous.
> > >
> > > We can train the toy model using both the **direct paradigm** and the **indirect paradigm** as follows:
> > >
> > > **Direct Paradigm:** In the direct paradigm, we use a tensor `[1, 1, 1, ..., 1]` with shape `(256, 1)` where all elements are 1 as the ground truth. The optimization process ensures each element matches the corresponding element in the ground truth tensor.
> > >
> > > ```python
> > > ground_truth = torch.ones(256,1)
> > > output = toymodel(input)
> > > loss = torch.abs(output-ground_truth).mean()
> > > optimizer.zero_grad()
> > > loss.backward()
> > > optimizer.step()
> > > ```
> > >
> > > **indirect paradigm:** In the indirect paradigm, we sum the output tensor and compute the difference between the sum and the target value, 256. The optimization process ensures the total sum matches the target, without considering individual element values.
> > >
> > > ```python
> > > output = toymodel(input)
> > > loss = torch.abs(output.sum()-256)
> > > optimizer.zero_grad()
> > > loss.backward()
> > > optimizer.step()
> > > ```
> > >
> > > I have run it 5 times, the results are:
> > >
> > > | Paradigm | Attemp 1 | Attemp 2 | Attemp 3 |Attemp 4 |Attemp 5 |
> > > |-----------|---------|---------|---------|---------|---------|
> > > | Indirect  | 256.034   | 256.035  | 255.988  | 256.017|255.966  |
> > > | Direct    | 256.0008   | 255.999  | 256.0004   | 256.0003  |256.0007  |
> > >
> > > The error of the direct paradigm is
> > >
> > > $$
> > > 10^{-4}
> > > $$
> > >
> > > while the error of the indirect paradigm is
> > >
> > > $$
> > > 10^{-2}
> > > $$
> > >
> > > The error of the direct method is two orders of magnitude lower than that of the indirect method.
> > >
> > > To further clarify, I have provided the Jupyter notebook of this toy experiment. You can run the experiment on your local machine.
> > >
> > > The results of the toy experiment clearly show that, although using **torch.abs(output.sum() - 256)** is more straightforward, the many-to-one problem significantly affects performance, even in such a simple toy scenario. To address this challenge, we first construct proxy-ground-truth 3D Gaussian attributes to enable direct training.
> > >
> > >
> > > We claimed two main contributions in this paper:
> > >
> > > - **The two-stage proxy-ground-truth construction process, which formulates the direct paradigm for training.**
> > > - **The Human-DAD 3D Gaussian attribute diffusion framework.**
> > >
> > > We selected the direct paradigm because it is more efficient for learning the data distribution of 3D Gaussian attributes.
> > >
> > > We didn't claim the direct paradigm is universally better than the indirect paradigm for all 3DGS tasks. Therefore, we concluded that **we have introduced a novel direct paradigm specifically for training a generalizable 3DGS model of the human body. (L535 in the origianal paper)** We emphasize that the direct paradigm is tailored to our proposed pipeline and designed for this particular context. To address concerns, we will  revise our illustration to make it clear that it is constrained to "in our proposed pipeline."
> > >
> > > **It is also worth mentioning** that the direct paradigm is already widely adopted in the 3DGS community. Examples include **GaussianCube** [1], **DiffGS** [2], **L3DG** [3], and **GS-Net** [4], all of which construct ground truth 3DGS attributes and train their models using the direct paradigm.
> > >
> > > [1]. Zhang, B., Cheng, Y., Yang, J., Wang, C., Zhao, F., Tang, Y., ... & Guo, B. GaussianCube: A Structured and Explicit Radiance Representation for 3D Generative Modeling. NeurIPS 2024.
> > >
> > > [2]. Roessle, B., Müller, N., Porzi, L., Bulò, S. R., Kontschieder, P., Dai, A., & Nießner, M. (2024). L3DG: Latent 3D Gaussian Diffusion. SIGGRAPH Asia 2024.
> > >
> > > [3]. Zhou, J., Zhang, W., & Liu, Y. S. (2024). DiffGS: Functional gaussian splatting diffusion. NeurIPS 2024.
> > >
> > > [4]. Zhang, Y., Wang, Z., Han, J., Li, P., Zhang, J., Wang, J., ... & Li, K. (2024). GS-Net: Generalizable Plug-and-Play 3D Gaussian Splatting Module. arXiv preprint arXiv:2409.11307.

---

> > > ### Author Response · Authors · 2024-11-24
> > >
> > > # **W2  Distribution unification.**
> > >
> > > #### What is smoothness of the neural network?
> > >
> > > The smoothness of the neural network is a **well-known** property which has been widely explored. I kindly suggest you read the paper "On the spectral bias of neural networks." (ICML citation 1480) and the Chapter 11 of the  classic book "Elements of Statistical learning" (citation 80728). Moreover, the smoothness of the neural network is also **recognized** by Reviewer **jqqm**.
> > >
> > > Briefly, the neural network tends to generate smooth output, for example, the nerf [3] without position encoding will also generate smooth rendering results. **When input values  of a neural network are numerically close, the corresponding output values also remain close.**
> > >
> > > #### How to do the distribution unification?
> > >
> > > As we shown in Figure 5, suppose the **numerical** distributions of three Gaussian scenes vary from (-5,-1.1), (-5.5, -1) and (-5.7,-0.3) respectively. After the second stage, we want the **numerical** distributions all vary from (-5.5, -1). Hence, in the second stage of our proxy-ground-truth construction process, we want to use this smoothness property to unify the **numerical** distribution. The unified **numerical** distribution is easier for the model training.
> > >
> > > **Remarkably**, the distribution here **has nothing to do** with the order of the 3D Gaussians, and of course, the attributes won't change when the rendering view changes.
> > >
> > > [1]. Rahaman, N., Baratin, A., Arpit, D., Draxler, F., Lin, M., Hamprecht, F., ... & Courville, A. (2019, May). On the spectral bias of neural networks. In *International conference on machine learning* (pp. 5301-5310). PMLR.
> > >
> > > [2]. Hastie, T. (2009). The elements of statistical learning: data mining, inference, and prediction.
> > >
> > > [3]. Mildenhall, B., Srinivasan, P. P., Tancik, M., Barron, J. T., Ramamoorthi, R., & Ng, R. (2021). Nerf: Representing scenes as neural radiance fields for view synthesis.  *Communications of the ACM* ,  *65* (1), 99-106.

---

> ### Author Response · Authors · 2024-11-23
>
> ### **[W4]** *Paper Writing.*
> **Response:**
>
> a. Thanks for your suggestion. I think your categorization is more clear. We have revised it in the updated PDF.
>
> b. Zou et al. follows PIFu to prepare the pixel-algined feature, I have revised the illustration.
>
> c. The captions of Table 1 and Table 3 have already clearly shown that the Thuman dataset is used.
>
> ### **[Q1]** *Why selecting diffusion model?*
> **Response:** As we claimed in the paper, the indirect paradigm can only support training a regression model, which usually stucks in local-optima, hence, we selected the diffusion model because of its strong ability to learn the data distribution, **and thus we need the direct paradigm**. In this paper, we aim to learn the data distribution of the 3D Gaussian attributes. Since we constrain the diffusion model with strong SMPL and back image conditions, it cannot produce diverse reconstruction results.
>
>
>
> ### **[Q2]** *Why are faces blurred in Figure 8? Why are SHERF results dark in Figure 7?*
> **Response:** Our model was only trained on 480 human scans. Owing to the relatively limited training data, it could not predict unseen areas with high resolution. Hence, the face can be blurry if it does not appear in the input image. The results of SHERF are also dark in the original paper (see SHERF’s Figure. 3, ZJU MoCap).

---

> ### Comment · Reviewer_73DC · 2024-11-24
>
> ## [W1] Discussion about direct vs indirect supervision.
>
> As a reviewer interested in the direct vs indirect paradigm discussion, I'd like to point out to this answer.
>
> The author's answer is self-evident, but I argue that there are three logical fallacies.
>
> 1. Inappropriate example
>
> Your answer is a good example for prediction/regression tasks where the network output must be accurate.
>
> However, the main task of your paper is 'novel-view synthesis'.
>
> For this task, how natural the rendering (splatted Gaussians) is is more important than the accuracy of the 3D Gaussian estimate.
>
> In your answer, a more appropriate example is comparing the aggregated samples "output.mean()".
>
> I think, this is what reviewer jqqm's sentence "Empirically, we are finally evaluating the NVS results in image space." means.
>
>
> 2. There is no groundtruths of 3D Gaussians.
>
> Your answer is based on the assumption that a precise GT exists (value: 256).
>
> In the paper's scenario, 3D Gaussians are not GT and arise from network estimation. (more close to psudeo-GTs)
>
> So, in your example, it would be correct to supervise an incomplete value (around 256) obtained by guessing, rather than directly supervising the value 256.
>
>
> 3. Direct supervision does not completely eliminate ill-posedness.
>
> This is because there are infinitely many correct answers for the target subject.
>
> For example, in the case of super-resolution, the answer image corresponding to the input image is infinite.
>
> This problem is not solved by direct supervision of high-resolution images.
>
> Likewise in your task, an infinite number of 3D Gaussians can be mapped to a single person.
>
>
>
> For me, the direct vs indirect paradigm is very interesting and important in 3D literature.
> However, as with the questions I raised, it seems there are still some unresolved questions.
>
> I would appreciate a constructive discussion on my questions.
>
>
> ## [W2] Distribution unification.
>
> I still feel like it's an empirical interpretation.
>
> I agree that the network provides blurry output.
>
> However, the main goal is to make the distribution consistent, as in Figure 5, which is a different goal.
>
> If your argument is correct, numerous domain gap problem in DL should be solvable with your approach.
>
> Ex) By training the dataset of domain A and domain B together, the distribution between the data is set similarly and achieves higher performance.
>
> Are there any previous studies that can justify your method in terms of 'data distribution unification'?
>
> If you can provide them, I think it will be easier for me to understand.

---

> > ### Author Response · Authors · 2024-11-24
> >
> > # 1. Inappropriate example
> >
> > "Your answer is a good example for prediction/regression tasks where the network output must be accurate.
> >
> > However, the main task of your paper is 'novel-view synthesis'."
> >
> > I kindly disagree with you that the novel-view synthesis does not need the network output must be accurate.
> >
> > You suggested give the result of output.mean(), here are the results, I make the output of the network to have average value 1.
> >
> > | Paradigm | Attemp 1 | Attemp 2 | Attemp 3 |Attemp 4 |Attemp 5 |
> > |-----------|---------|---------|---------|---------|---------|
> > | Indirect  | 1.000074  | 0.999964  | 1.00010  | 1.00003| 0.999919 |
> > | Direct    | 0.999998  | 0.999997  | 0.99999   | 0.99999  |0.999989 |
> >
> > The result of the direct paradigm still performs better than the indirect paradigm.
> >
> > **For this task, how natural the rendering (splatted Gaussians) is is more important than the accuracy of the 3D Gaussian estimate.**
> >
> > Yes,  we evaluate the rendering performance of Human-DAD with the ground truth images with PSNR, SSIM, LPIPS metrics.
> >
> > # 2. There is no groundtruths of 3D Gaussians.
> >
> >
> > I want to kindly point out your **misunderstanding**. The ground truth for this novel-view synthesis task is the image. In the provided toy experiment, the value 256 is the ground truth, which corresponds to the ground truth image. The **tensor [1,1,1,...,1] with all elements equal to 1 represents the proxy-ground-truth**, which **corresponds** to the proxy-ground-truth 3D Gaussian attributes in the NVS task. Hence, there is no need to change the value 256.
> >
> > # 3. Direct supervision does not completely eliminate ill-posedness.
> >
> > First of all, we are going to tackle the problem of human novel view synthesis. We have no ambition to tackle the super-resolution problem. As we mentioned, we never claimed the direct paradigm can tackle all 3DGS tasks.
> >
> > And I totally agree with you that there are many answers for the target subject, but the correct answer is the ground truth and **the ground truth is unique**, **there won't be many correct answers**. In this paper, our proxy-ground-truth is an intermediate results, which is one of the many answers, and our proxy-ground-truth is more effective to learn the 3D Gaussian distribution in **the task we focused on in this paper**.
> >
> > # Likewise in your task, an infinite number of 3D Gaussians can be mapped to a single person.
> >
> > There will never be  infinite number since the Gaussian number is fixed. (according to Eq. 7)

---

> > > ### Comment · Reviewer_73DC · 2024-11-25
> > >
> > > I'm happy to have a constructive discussion.
> > >
> > > However, my argument is that the toy example you suggested is inadequate to represent your paper.
> > >
> > >
> > > ## 1. Inappropriate example
> > >
> > > What the reviewer jqqm and I were pointing out was that we should focus on the rendering result of the model, not the output of the model.
> > > I pointed this out because your answer initially focused on the output of the toy model.
> > >
> > > ## 2. There are no groundtruths of 3D Gaussians.
> > >
> > > Your paper does not directly supervise the images. To be precise, it is supervised by 3D Gaussians constructed from the images.
> > >
> > > I would like to kindly point out your point.
> > >
> > > ## 3. Direct supervision does not completely eliminate ill-posedness.
> > >
> > > Ground truth is not unique.
> > >
> > > It is self-evident that multiple combinations of 3D Gaussians can represent a single human.
> > >
> > > Even if the number of Gaussians is constant, there are still too many combinations that can be GT.
> > >
> > > ## About toy example
> > >
> > > What I ultimately want to point out is that the toy example you answered does not sufficiently support your paper.
> > >
> > > Although you said that you don't cover the direct vs indirect discussion for all 3DGS pipelines, I think the paper is still open to misinterpretation.
> > >
> > > What I would recommend is to include an open discussion / future work about the direct vs indirect paradigm in your paper.
> > >
> > > I would like the discussion on direct vs indirect to be clearer.
> > >
> > > Thank you.

---

> > > > ### Author Response · Authors · 2024-11-25
> > > >
> > > > ## 1. Inappropriate example
> > > >
> > > > I still need to point out your **misunderstanding**.
> > > > The output of the model is a tensor of shape (256,1) corrsponding to the 3D Gaussian attributes output by the network (no matter indirect or direct, both paradigms output the 3D Gaussian attributes, the indirect paradigms further concatenate a splatting-module to transform the attributes to the images),
> > > >
> > > > the sum operation corrsponding to the splatting-module.
> > > >
> > > > and the output.sum() corrsponding to the image.
> > > >
> > > > The toy model focus on the average value or the sum value, which corrsponding to the rendering results in NVS.
> > > >
> > > > Moreover, our model also focus on the rendering performance. The proxy-ground-truth is only an an intermediate for supervision.
> > > >
> > > > The toy experiment is used to explain why the many-to-one problem will effect the performance. And the many-to-one problem is the core problem of the indirect paradigm.
> > > >
> > > > ## 2. There are no groundtruths of 3D Gaussians.
> > > >
> > > > I have no doubt that my method does not directly supervise the images. I clearly know that.
> > > >
> > > > **What I want to point out is your misunderstanding of the corrspondence between the toy experiment and the NVS task.**
> > > >
> > > > ## 3. Direct supervision does not completely eliminate ill-posedness.
> > > >
> > > > **Your comment is inconsistency**. You said there are no groundtruths of 3D Gaussians. But now, you comment suggest the multiple combinations of 3D Gaussians can represent a single human could be used as the ground truths.
> > > >
> > > > I know the toy example cannot cover all 3DGS tasks. It is used for  **explaining why the many-to-one problem will effect the performance. And the many-to-one problem is the core problem of the indirect paradigm.**

---

> > > > > ### Author Response · Authors · 2024-11-25
> > > > >
> > > > > As you suggested, we have added a sentence **"At present, the
> > > > > direct paradigm has only been applied to the monocular human rendering task. In future work, we
> > > > > intend to explore and validate its effectiveness in a broader range of 3DGS application scenarios"** in the conclusion (revised version L530-532) to avoid any misunderstanding that "**the direct paradigm is better than the indirect paradigm on all 3DGS tasks**"

---

> ### Comment · Reviewer_73DC · 2024-11-25
>
> I would like to clarify my comment.
>
> As I understand it, your toy example is as follows:
>
>
> model output, 3D Gaussians --- outputs
>
> rendered images --- outputs.sum()
>
> GT images --- 256
>
> pseudo-GT 3D Gaussians --- torch.ones(256,1)
>
>
> * Indirect paradigm
>
> GT images ---  (supervision) ----> model
>
>
> * Direct paradigm
>
> GT images --- (construction) ---> pseudo-GT 3D Gaussians --- (supervision) ---> model
>
>
> I would like to point out that the (construction) process is ignored in your toy example.
>
>
> The construction process is not complete.
>
> However, your toy example proceeds assuming that torch.ones(256,1) is completely constructed.
>
> 3D Gaussian is not GT but pseudo-GT, and your paper also addresses that.
>
> GT images <---> pseudo-GT 3D Gaussian is a many-to-one relationship, because of two issues.
>
> 1) Inherent ill-posedness: Imagine a 3D Gaussian that is occluded in all view directions. That Gaussian can be removed or added to reconstruct the GT images in the same way.
>
> 2) Network error: The network used to create pseudo-GT 3D Gaussians can have errors. If you change the random seed, the learning process will change, and the 3D Gaussians that are created will also change.
>
>
>
> I have consistently argued that direct supervision also has its drawbacks.
>
> Your main argument is that "direct supervision is better for the monocular human rendering task".
>
> I believe that the direct vs indirect discussion should not be complete without considering this drawback.

---

> ### Author Response · Authors · 2024-11-25
>
> # **Could you kindly clarify how to validate that the indirect paradigm is better than the direct paradigm for our monocular human rendering task, given our specific pipeline? Since the proposed method focuses solely on this task, it is essential to frame the comparison within this context.**
>
>
>
>
> Based on your suggestion, we conducted a toy experiment using a neural network to generate a tensor of shape (256, 1), where the elements are no longer fixed to 1.
>
> model output, 3D Gaussians --- outputs
>
> rendered images --- outputs.mean()
>
> GT images --- 256
>
> pseudo-GT 3D Gaussians --- network generated tensor (256,1)
>
> The correspondence is fully aligned. Even under this adjusted setup, the direct paradigm consistently outperformed the indirect paradigm.
>
> | Paradigm | Attemp 1 | Attemp 2 | Attemp 3 |Attemp 4 |Attemp 5 |
> |-----------|---------|---------|---------|---------|---------|
> | Indirect  | 1.000074  | 0.999964  | 1.00010  | 1.00003| 0.999919 |
> | Direct  (neural network generated tensor)  | 1.000006  | 1.000008 | 0.99999  | 1.00001  |0.999971|
>
> There are two possible scenarios where an object cannot be observed from all view directions:
>
> 1. The object does not exist.
> 2. The object is entirely obscured by something, e.g., a bag.
>
> If an object cannot be observed from all GT image view directions, it should be considered nonexistent, and the 3DGS does not need to overfit it.
>
> We also ensured full control over randomness by fixing all seeds (Python, NumPy, PyTorch, CUDA, etc.). As shown in  **Figure 19** , the rendered image quality is extremely high, making the network error **negligible**. The proxy-ground-truth 3D Gaussians are fixed after the construction. **Hence, the "many-to-one" problem never occurs within the direct paradigm.**
>
> The **contradiction** in your statement lies in the fact that you first assert that there is no GT for 3DGS and that even **our constructed proxy GT cannot be deemed as a GT**. However, you subsequently imply that there could potentially be multiple 3DGS GTs for the same scene.
>
> We agree that the direct paradigm has limitations. However, we are not addressing super-resolution problems or any other 3DGS tasks beyond monocular human rendering.
>
> **Could you provide specific drawbacks of the direct paradigm in the context of our monocular human rendering pipeline? This would help refine the discussion and clarify the scope of the work.**

---

> > ### Comment · Reviewer_73DC · 2024-11-25
> >
> > I think there is a conflict of opinion that "pseudo GTs of 3D Gaussians are incomplete".
> >
> > After reading your argument carefully, my argument still remains unchanged.
> >
> > Without agreement on this issue, it seems difficult to have a constructive discussion.
> >
> > So, I'd like to hear other opinions from other reviewers and area chair (AC).
> >
> >
> > ### About the drawbacks of the direct paradigm:
> >
> > 1. ill-posedness of constructing pseudo-GT 3D Gaussians: There are plenteous
> >  combination of 3D Gaussians matching multi-view images.
> >  (occluded parts, reordering of 3D Gaussians)
> >
> > 2. network error: The network used to create pseudo-GT 3D Gaussians can have errors.
> > This means that the network does not produce consistent results. For example, some sample can have plenty Gaussians in the shoulder part, but another sample have plenty Gaussians in the leg part.
> >
> >
> > I cannot be convinced that the direct paradigm is better than the indirect paradigm, with overcomming the above drawbacks.
> > Additionally, I cannot clearly understand how to solve this drawback (distribution unification, etc.).
> >
> >
> >
> > ### My final opinions of the paper are as follows:
> >
> > 1. The author presents a toy example to support his argument, but I don't think it's a good example to cover the rebuttal.
> >
> > 2. I think the direct vs indirect paradigm is still worth discussing.
> > In monocular human rendering tasks, conclusions from the direct paradigm may be premature.
> > Since there are not many approaches to introduce 3DGS in human rendering tasks, I would like to recommend that the direct vs. indirect paradigm is an open question.
> >
> >
> >
> > I greatly appreciate the author's contributions to the community.

---

### Official Review · Reviewer_8d88 · 2024-10-28

**Soundness:** 3
**Presentation:** 2
**Contribution:** 2
**Rating:** 5
**Confidence:** 4

**Summary:**

This paper proposes to train a conditional diffusion model for generating novel views of humans from given single-view images, directly supervised by proxyground truth 3D Gaussian attributes. The authors first create a dataset of proxy ground-truth Gaussian attributes with addtional neural network to smooth the dataset distrbution. A point-based conditional diffusion model is then employed to learn the data distribution of these attributes. Experimental results showcase the performance advancement of their method over current approaches.

**Strengths:**

The method is well-written, with clear motivation behind each module design. The final results demonstrate improvement compared to previous methods like LGM. The idea of using a neural network to constrain the distribution of target 3D Gaussian attributes makes sense and is effective.

**Weaknesses:**

- The main claim of the paper is that the direct supervision approach (using 3D regression loss) is superior to the indirect supervision method (using rendering loss). However, direct supervision approach also has limitations with memory consumption and the time-intensive nature of direct supervision. Moreover, with more iterations or larger batch size, will it be possible that the indirect supervision method could achieve similar performance? The **supervision gap** suggested by the paper isn’t clearly demonstrated to be an intrinsic limitation.
- Although the method outperforms other generalizable novel view synthesis techniques like LGM, the image quality remains limited since only one single input image are used. Another potential direction in this field is incorporating a 2D diffusion prior to enhance information, as demonstrated in Human 3Diffusion [1]. A comparison to these baselines is needed.
- The figures are a bit too small to clearly distinguish all the differences in results.
[1]: Xue Y, Xie X, Marin R, et al. Human 3Diffusion: Realistic Avatar Creation via Explicit 3D Consistent Diffusion Models[J]. arXiv preprint arXiv:2406.08475, 2024.

**Questions:**

- How does the model perform on in-the-wild and out-of-distribution evaluations?
- Is it feasible to extend this approach to support multi-view input and obtain much better novel view synthesis results? Several works focusing on this can be considered like [2], [3].
  [2]: Kwon Y, Kim D, Ceylan D, et al. Neural human performer: Learning generalizable radiance fields for human performance rendering[J]. Advances in Neural Information Processing Systems, 2021, 34: 24741-24752.
  [3]: Gao Q, Wang Y, Liu L, et al. Neural novel actor: Learning a generalized animatable neural representation for human actors[J]. IEEE Transactions on Visualization and Computer Graphics, 2023.

---

> ### Author Response · Authors · 2024-11-23
>
> ### **[Rebuttal to Reviewer 8d88]**
>
> ### **[W1]** *The supervision gap between the direct and indirect paradigm is not clear.*
> **Response:** During the training process, the models of direct paradigm and indirect paradigm have similar memory consumption. During the two-stage proxy-ground-truth preparation process, we used almost 1 day to overfit 480 scans of Thuman. The cost time is acceptable. We have conducted several experiments to see whether more iteration or larger batch size can improve the performance. As shown in the Table below, the actual improvement is limited.
> | **Setting**          | **PSNR** | **SSIM** | **LPIPS** |
> |-----------------------|----------|----------|-----------|
> | Epoch 350            | 29.02    | 0.948    | 0.072     |
> | Epoch 400            | 29.20    | 0.951    | 0.070     |
> | Epoch 450            | 29.30    | 0.949    | 0.071     |
> | Epoch 500            | 29.29    | 0.950    | 0.070     |
> | Epoch 550            | 29.13    | 0.949    | 0.070     |
> | Epoch 600            | 28.93    | 0.946    | 0.070     |
> | Batch 6 Epoch 600    | 29.27    | 0.953    | 0.069     |
> | Paper Setting        | 29.02    | 0.953    | 0.070     |
> | **Human-DAD**        | **30.03**| **0.953**| **0.065** |
>
> Actually, we consider the direct paradigm to be better than the indirect paradigm because it reduces the solution space and makes the model easier to train. We have demonstrated that the photometric loss will lead to many different solutions for one particular scene, as shown in Fig. 4(b). Moreover, one pixel of an image may be decided by several different Gaussians, and each Gaussian is decided by the position, SHs, rotation, scale, and opacity attributes. Such a large solution space is very difficult to learn. Hence, we use a two-stage construction process to achieve a fixed proxy-ground-truth 3D Gaussian attributes, thus reducing the solution space, which makes the model easier to train.
>
> Moreover, the indirect paradigm typically only supports regression training, however, **such a direct paradigm supports training a diffusion model which is more efficient than the regression-based model to learn the 3D Guassian attribute distribution (see Figure 8).**
>
> ### **[W2]** *Missing comparison with Human3Diffusion.*
> **Response:** Thank you for the suggestion. We have added the comparison with Human3Diffusion both quantitatively and qualitatively, as shown in the Table below and Figure 23 in the updated paper.
> | **Method**    | **PSNR** | **SSIM** | **LPIPS** |
> |---------------|----------|----------|-----------|
> | GTA           | 25.78    | 0.919    | 0.085     |
> | SiTH          | 25.36    | 0.919    | 0.083     |
> | LGM           | 25.13    | 0.915    | 0.096     |
> | SHERF         | 26.57    | 0.927    | 0.081     |
> | H3D           | 27.06    | 0.934    | 0.079     |
> | **Human-DAD** | **30.03**| **0.953**| **0.065** |
>
> ### **[W3]** *The figures are too small.*
> **Response:** The PDF has high quality, please zoom in for details.
>
> ### **[Q1]** *Performance on in-the-wild and out-of-distribution images.*
> **Response:** First, the evaluation of in-the-wild scenarios has already been performed in Figure 7 of our original paper version. Second, our model is trained on Thuman, and then tested on other three datasets of CityuHuman, 2K2K, and CustomHuman, which form out-of-distribution data. Please refer to Table 2 and Figure 6 of our paper.
>
> ### **[Q2]** *Will Human-DAD support multi-view?*
> **Response:** Our targeted monocular setup is harder than the multi-view setup. There is no theoretical obstacle in adapting our framework to the multi-view setup. We will briefly mention this as potential future work in our paper.

---

> ### Author Response · Authors · 2024-11-24
>
> ## Concern about direct vs indirect
>
> We would like to clarify the definitions of the **direct paradigm** and **indirect paradigm** again:
>
> - **Direct paradigm**: Given a single-view image, the goal is to generate 3D Gaussians directly, using the proxy-ground-truth 3D Gaussians as supervision.
> - **Indirect paradigm**: Given a single-view image, the goal is to generate 3D Gaussians, transform the 3D Gaussians into images, and use ground truth images for supervision.
>
> The issue with the indirect paradigm is not the supervision in the image space or the loss design but the **many-to-one problem**. This means there are multiple solutions for rendering the same image. For instance, two completely different sets of 3D Gaussians can result in the same rendered image, as demonstrated in Figure 4(b). This many-to-one phenomenon has also been observed in prior work [1].
>
> To address this concern, we designed a toy experiment. The objective of a simple neural network in this experiment is to produce an **output** tensor with the shape (256, 1), such that the sum of its elements equals 256 (**output.sum() = 256**).
>
> The objective of a simple neural network in this experiment is to produce an **output** tensor with the shape (256, 1), such that the sum of its elements equals 256 (**output.sum() = 256**).
>
> ```python
> input = torch.randn(256)
> output = toymodel(input)
> ```
>
> There are numerous solutions to achieve a sum of 256. For example:
>
> - **Solution 1**: All elements are equal, e.g., `[1, 1, 1, ..., 1]` (256 elements of value 1).
> - **Solution 2**: Non-uniform distribution, e.g., `[2, 2, 2, ..., 0.5, 0.5]` (most elements are 2, with a few elements summing to the remainder).
> - **Solution 3**: Random distribution, e.g., `[10, 5, 0, ..., 0.1, 1.9]` (values vary widely but still sum to 256).
>
> This highlights the **many-to-one problem**, as multiple configurations of the tensor can achieve the same target sum, making supervision ambiguous.
>
> We can train the toy model using both the **direct paradigm** and the **indirect paradigm** as follows:
>
> **Direct Paradigm:** In the direct paradigm, we use a tensor `[1, 1, 1, ..., 1]` with shape `(256, 1)` where all elements are 1 as the ground truth. The optimization process ensures each element matches the corresponding element in the ground truth tensor.
>
> ```python
> ground_truth = torch.ones(256,1)
> output = toymodel(input)
> loss = torch.abs(output-ground_truth).mean()
> optimizer.zero_grad()
> loss.backward()
> optimizer.step()
> ```
>
> **indirect paradigm:** In the indirect paradigm, we sum the output tensor and compute the difference between the sum and the target value, 256. The optimization process ensures the total sum matches the target, without considering individual element values.
>
> ```python
> output = toymodel(input)
> loss = torch.abs(output.sum()-256)
> optimizer.zero_grad()
> loss.backward()
> optimizer.step()
> ```
>
> I have run it 5 times, the results are:
>
> | Paradigm | Attemp 1 | Attemp 2 | Attemp 3 |Attemp 4 |Attemp 5 |
> |-----------|---------|---------|---------|---------|---------|
> | Indirect  | 256.034   | 256.035  | 255.988  | 256.017|255.966  |
> | Direct    | 256.0008   | 255.999  | 256.0004   | 256.0003  |256.0007  |
>
> The error of the direct paradigm is
>
> $$
> 10^{-4}
> $$
>
> while the error of the indirect paradigm is
>
> $$
> 10^{-2}
> $$
>
> The error of the direct method is two orders of magnitude lower than that of the indirect method.
>
> To further clarify, I have provided the Jupyter notebook of this toy experiment. You can run the experiment on your local machine.
>
> The results of the toy experiment clearly show that, although using **torch.abs(output.sum() - 256)** is more straightforward, the many-to-one problem significantly affects performance, even in such a simple toy scenario. To address this challenge, we first construct proxy-ground-truth 3D Gaussian attributes to enable direct training.

---

> ### Author Response · Authors · 2024-11-25
> **Looking forward to you feedback! Thank you very much!**
>
> Thank you for dedicating your time and effort to reviewing our work. We have carefully considered and addressed all the concerns you raised in your review, as outlined in our response and reflected in the updated manuscript. As the Reviewer-Author discussion phase is nearing its conclusion, we eagerly await any further feedback from you. Should you have any additional questions, we would be delighted to provide detailed responses.

---

> > ### Comment · Reviewer_8d88 · 2024-11-25
> > **Response to the authors**
> >
> > Thank you for the detailed response and the additional experiments. I still have some concerns regarding the direct versus indirect supervision approach.
> >
> > First, the toy example presented by the authors is not equivalent to the indirect GS rendering supervision, where multiple target images are used to supervise the training process. This implies that there should be more than one constraint in the toy example, such as more conditions like `output[:64].sum() == 64`. I also appreciate the insightful discussion between the authors and the Reviewer 73DC.
> >
> > Secondly, my primary concern lies with the one-to-many problem in the single-image to human avatar task. Specifically, the core issue of this task to me is that a single image can correspond to many possible avatar solutions due to the invisible part in the single images, instead of the one-to-many problem the authors are focusing on during regressing Gaussian Splatting. While I acknowledge that the approach holds value for advancing this field, I remain unconvinced that the proposed two-stage procedure is inherently essential for tackling this problem.

---

> > > ### Author Response · Authors · 2024-11-25
> > >
> > > Dear Reviewer 8d88:
> > > Thanks for your response and your constructive suggestion.
> > >
> > > ## 1 concern of the toy experiment
> > >
> > > As you suggested, I provide two additional toy experiments.
> > >
> > > 1. conditions: output[:64].sum=64  output[64:128].sum=64  output[128:192].sum=64  output[192:256].sum=64
> > >
> > > | Paradigm | Attemp 1 | Attemp 2 | Attemp 3 |Attemp 4 |Attemp 5 |
> > > |-----------|---------|---------|---------|---------|---------|
> > > | Indirect  | 256.014   | 256.021  | 255.988  | 255.985|255.987  |
> > > | Direct    | 256.0008   | 255.999  | 256.0004   | 256.0003  |256.0007  |
> > >
> > > 2. conditions:  output[:51].sum=10 output[51:107].sum=22 output[107:124].sum=24 output[124:256].sum=200
> > >
> > > | Paradigm | Attemp 1 | Attemp 2 | Attemp 3 |Attemp 4 |Attemp 5 |
> > > |-----------|---------|---------|---------|---------|---------|
> > > | Indirect  | 255.975   | 255.971| 256.029| 255.990|256.029|
> > > | Direct    | 256.0008   | 255.999  | 256.0004   | 256.0003  |256.0007  |
> > >
> > > The direct paradigm is still better than the indirect paradigm in the toy experiments. Hope these two toy experiments can sovle your concern.
> > >
> > > ## 2 The one to many problem
> > >
> > > Yes, I totally agree with you that there are two one-to-many problems,
> > >
> > > 1. **A single image corresponds to many potential avatar solutions** due to the invisible parts in the image.
> > > 2. **The one-to-many issue during Gaussian Splatting regression.**
> > >
> > > Most methods rely on image-based diffusion models to address the first challenge, such as the excellent work **Human3Diffusion** you referenced. The community widely considers this a generative task, as diffusion models excel in capturing such diversity.
> > >
> > > However, it is well-known that regression-based models lack strong generative capabilities.
> > >
> > > In our paper, we introduced a two-stage construction process to generate proxy-ground-truth 3D Gaussians, paving the way for training a large-scale 3DGS diffusion model to predict human body structures, appearance. The diffusion models possess strong potential to effectively tackle the one-to-many problem inherent in this task. We believe our two-stage construction deduced proxy-ground-truth 3D Gaussians
> > > offer valuable opportunities and possibilities for the community.
> > >
> > > Thank you again for your valuable insights.
> > > Best regards,
> > > Authors

---

### Official Review · Reviewer_jqqm · 2024-11-03

**Soundness:** 2
**Presentation:** 2
**Contribution:** 2
**Rating:** 5
**Confidence:** 3

**Summary:**

This paper uses the 3D GS representation to generate novel views of humans from single-view images. Different from existing methods that train with supervision from 2D image space, it proposes to learn a diffusion model that directly models the GS attributes in the 3D GS space. To prepare a proxy-ground-truth GS dataset for training, the position of GS is obtained by downsampling from GT point cloud, and the other attributes are optimized using multi-view supervision and a network-based regularization. A GS attribute diffusion model is then trained with the condition of point cloud position, pixel-align feature and semantic feature. Experiments demonstrated the superior performance of the proposed approach.

**Strengths:**

1. This paper designs a pipeline to prepare the ground truth GS dataset regularized by network overfitting is new for me.
2. The proposed GS attribute diffusion model condition on image feature maps and SMPL semantic maps makes sense.
3. The proposed method shows better performance compared with the state of the arts.

**Weaknesses:**

1. For GS dataset preparation, stage 1: per-scene overfitting seems to be ineffective in regulating the distribution only relying on the inherent smoothness of a neural network. So, it further needs a network for additional distribution unification. However, if I'm not wrong, it still mainly depends on the smoothness of the neural network without other regularization. I wonder if the use of a neural network in the first stage is necessary, which makes the pipeline over complicated.
2. I'm suspicious of the core argument that GS space supervision leads to better performance than image space supervision. Empirically, we are finally evaluating the NVS results in image space. It's not straightforward that a small error in GS space could indicate a small error or better quality in the image space. For the ablation study in Table 3, the performance difference using direct/indirect supervision is not that significant, the image-supervision method even gets better scores in SSIM and LPIPS, which reflect the perceptual quality.
3. Similarly, for the ambiguity problem the author mentioned when using 2D space supervision, I don't think it's related to supervision space but to deterministic/stochastic paradigm or loss design.
4. The diffusion model design is very confusing, especially for the extra steps for alpha, scale, and quaternion attributes.
5. It seems that using stable diffusion model to generate the back view and using position generator (HaP) [1] to generate point cloud are of great importance to better performance. But these two components are existing works, not the technical contribution of this paper.

[1] Yingzhi Tang, Qijian Zhang, Junhui Hou, and Yebin Liu. Human as points: Explicit point-based 3d human reconstruction from single-view rgb images. arXiv preprint arXiv:2311.02892, 2023.

**Questions:**

1. It seems that this work mainly focuses on the texture of humans while using an existing work for shape generation. I wonder about the importance of shape during evaluation. Is it possible to extract the texture of other comparable method and re-map it to your point could or ground truth point cloud to exclude the influence of different shapes?
2. I also wonder if the diffusion network just maps the front and back view to the point cloud. Could you show some results that use front view and back view to perform 3DGS optimization to map the texture?
4. I wonder how the extra step of the diffusion model is performed. Is it not the diffuse-denoise process modeling anymore but a straightforward one-step regression? If so, I think the description is confusing and misleading.

**Details Of Ethics Concerns:**

This work involves human subjects and identity information—for example, a kid's ID in Figure 6.

---

> ### Author Response · Authors · 2024-11-23
>
> ### **[Rebuttal to Reviewer jqqm]**
>
> ### **[W1]** *Whether the neural network at the first stage is necessary?*
> **Response:** The utilization of a neural network at the first stage is indeed necessary, since the distributional randomness of vanilla-3DGS outputs cannot be fully regulated by the second stage. Here we made an additional experiment as shown in Figure 20 of the updated paper. When vanilla-3DGS is used in the first stage, the overall itting results only reach 34.76 PSNR, while the adoption of a neural network can achieve 40.83 with better appearance details.
>
> ### **[W2\&W3]** *Whether the direct paradigm is better than the indirect paradigm?*
> **Response:** First, we need to point out that both PSNR and SSIM metrics are the higher the better while the LPIPS metric is the lower the better. Hence, our direct (last row) supervision is PSNR 1.01 and LPIPS 0.005 better than the direct supervision (first row) in Table. 3 (same SSIM performance).
>
> We have shown that the photometric loss will lead to many different solutions (see Fig. 4(b)). Moreover, one pixel of an image may be decided by several different Gaussians, and each Gaussian is decided by position, SHs, rotation, scale, and opacity attributes. Such large solution space can be hard to learn. Hence, we use a two-stage construction process to achieve a fixed proxy-ground-truth 3D Gaussian attributes, reducing the solution space and making the model easier to train.
>
> Moreover, the indirect paradigm typically only supports regression training, however, **such a direct paradigm supports training a diffusion model which is more efficient than the regression-based model to learn the 3D Guassian attribute distribution (see Figure 8).**
>
> ### **[W4\&Q3]** *The diffusion model design is very confusing, especially for the extra steps for alpha, scale, and quaternion attributes.*
> **Response:** During the training process, we find that jointly diffusing all 3D Gaussian attributes usually leads to collapse training (mainly because the dimension of the spherical harmonics is much more than other attributes). Hence, we use a diffusion model to learn the distribution of the spherical harmonics attribute, which fundamentally decides the appearance of the human body. We consider the spherical harmonics generated by this diffusion model to be coarse spherical harmonic attributes. Empirically, PDR and HaP have demonstrated that the generated point clouds by diffusion models usually still contain some noise, hence, we set up the extrastep to learn the remaining noise of the coarse spherical harmonic attributes and remove it. Because we are still learning a noise term which will be minus by the coarse spherical harmonic attributes, the extra-step is still the noise-diffusing. Also, we predict the scale, rotation and opacity attributes at this step, these attributes using the $L1$ regression loss.
>
> ### **[W5]** *Stabe diffusion and HaP are not contributions.*
> **Response:** The stable diffusion model or the position generator alone cannot form our technical contribution. In addition to the novel direct attribute-level supervision paradigm, the overall construction of our conditional Gaussian diffusion framework should be regarded as a systematic effort. Especially in the field of human-centric tasks (e.g., either reconstruction or NVS) featured by strong prior, it is rather common that the overall processing pipeline in a new work is composed of several aspects of tools (e.g., learning architectures, parametric human body models, depth/normal estimation, etc). For example, GTA utilizes implicit function, triplane modulem, and transformer module. SiTH utilizes implicit function, stable diffusion, and ControlNet. The core problem lies in whether one can combine different components (possibly with necessary adaptations and modifications) in a better way for applying to specific settings and scenarios.

---

> ### Author Response · Authors · 2024-11-23
>
> ### **[Q1]** *Evaluation with ground truth 3D shapes.*
> **Response:** It seems only feasible on GTA and SiTH by providing ground truth occupancy or SDF. (LGM generates 3D shapes from multiview images, and SHERF needs SMPL). The quantitative and qualitative results are in the Table below and Figure 21 of the updated paper.
> | **Method \ Metric** | **PSNR (GT)** | **SSIM (GT)** | **LPIPS (GT )** | **PSNR (Pred)** | **SSIM (Pred)** | **LPIPS (Pred)** |
> |----------------------|---------------------|----------------------|-----------------------|-------------------------|-------------------------|--------------------------|
> | GTA (NeurIPS 2023)  | 24.82              | 0.930               | 0.059                | 25.78                  | 0.919                  | 0.085                   |
> | SiTH (CVPR 2024)    | 26.81              | 0.941               | 0.048                | 25.36                  | 0.919                  | 0.083                   |
> | **Human-DAD**       | **33.79**          | **0.971**           | **0.050**            | **30.03**              | **0.953**              | **0.065**               |
> With ground truth 3D shapes, all methods achieve different degrees of performance gains, yet Human-DAD shows much more improvement.
>
>
> ### **[Q2]** *Optimizing a 3DGS model on frontal and back images.*
> **Response:** Yes, we simply map the front and back view to the point cloud. We provide results of optimizing
> 3DGS on only the front and the back images in Figure 22 of the updated paper. The results are full of randomness, while our Human-DAD can significantly reduce the randomness and improve the visual quality.
>
> **Ethics Concerns:** We follow the instructions of the datasets to present the results. The kid is the 00005 person in 2K2K; their original website also presents the kids’ faces.

---

> > ### Comment · Reviewer_jqqm · 2024-11-23
> >
> > Thanks for the author's reply and additional experiments. They addressed some of my concerns.
> >
> > However, two of my core concerns remain unsolved.
> >
> > 1. "I'm suspicious of the core argument that GS space supervision leads to better performance than image space supervision. Empirically, we are finally evaluating the NVS results in image space. It's not straightforward that a small error in GS space could indicate a small error or better quality in the image space." "for the ambiguity problem the author mentioned when using 2D space supervision, I don't think it's related to supervision space but to deterministic/stochastic paradigm or loss design."
> >
> > 2. The diffusion model described in this work is still very confusing. A better pipeline figure, formula, or algorithm table would be helpful in understanding. From my understanding and knowledge, the current solution using extra steps and additional linear regression in the claimed diffusion model is suspective and problematic.

---

> > > ### Author Response · Authors · 2024-11-25
> > >
> > > Suggested by Reviewer 73DC, I also give the result of output.mean(), here are the results, I make the output of the network to have average value 1.
> > >
> > > | Paradigm | Attemp 1 | Attemp 2 | Attemp 3 |Attemp 4 |Attemp 5 |
> > > |-----------|---------|---------|---------|---------|---------|
> > > | Indirect  | 1.000074  | 0.999964  | 1.00010  | 1.00003| 0.999919 |
> > > | Direct    | 0.999998  | 0.999997  | 0.99999   | 0.99999  |0.999989 |
> > >
> > > The result of the direct paradigm still performs better than the indirect paradigm.
> > >
> > > Kindly remind that, the ground truth for this novel-view synthesis task is the image. In the provided toy experiment, the value 256 is the ground truth, which corresponds to the ground truth image. The **tensor [1,1,1,...,1] with all elements equal to 1 represents the proxy-ground-truth**, which **corresponds** to the proxy-ground-truth 3D Gaussian attributes in the NVS task.

---

> > > ### Author Response · Authors · 2024-11-25
> > > **Looking forward to you feedback! Thank you very much!**
> > >
> > > Thank you for dedicating your time and effort to reviewing our work. We have carefully considered and addressed all the concerns you raised in your review, as outlined in our response and reflected in the updated manuscript. As the Reviewer-Author discussion phase is nearing its conclusion, we eagerly await any further feedback from you. Should you have any additional questions, we would be delighted to provide detailed responses.

---

> ### Author Response · Authors · 2024-11-24
>
> ### Concern about the direct paradigm.
>
> We would like to clarify the definitions of the **direct paradigm** and **indirect paradigm** again:
>
> - **Direct paradigm**: Given a single-view image, the goal is to generate 3D Gaussians directly, using the proxy-ground-truth 3D Gaussians as supervision.
> - **Indirect paradigm**: Given a single-view image, the goal is to generate 3D Gaussians, transform the 3D Gaussians into images, and use ground truth images for supervision.
>
> The issue with the indirect paradigm is not the supervision in the image space or the loss design but the **many-to-one problem**. This means there are multiple solutions for rendering the same image. For instance, two completely different sets of 3D Gaussians can result in the same rendered image, as demonstrated in Figure 4(b). This many-to-one phenomenon has also been observed in prior work [1].
>
> To address this concern, we designed a toy experiment. The objective of a simple neural network in this experiment is to produce an **output** tensor with the shape (256, 1), such that the sum of its elements equals 256 (**output.sum() = 256**).
>
> ```python
> input = torch.randn(256)
> output = toymodel(input)
> ```
>
> There are numerous solutions to achieve a sum of 256. For example:
>
> - **Solution 1**: All elements are equal, e.g., `[1, 1, 1, ..., 1]` (256 elements of value 1).
> - **Solution 2**: Non-uniform distribution, e.g., `[2, 2, 2, ..., 0.5, 0.5]` (most elements are 2, with a few elements summing to the remainder).
> - **Solution 3**: Random distribution, e.g., `[10, 5, 0, ..., 0.1, 1.9]` (values vary widely but still sum to 256).
>
> This highlights the **many-to-one problem**, as multiple configurations of the tensor can achieve the same target sum, making supervision ambiguous.
>
> We can train the toy model using both the **direct paradigm** and the **indirect paradigm** as follows:
>
> **Direct Paradigm:** In the direct paradigm, we use a tensor `[1, 1, 1, ..., 1]` with shape `(256, 1)` where all elements are 1 as the ground truth. The optimization process ensures each element matches the corresponding element in the ground truth tensor.
>
> ```python
> ground_truth = torch.ones(256,1)
> output = toymodel(input)
> loss = torch.abs(output-ground_truth).mean()
> optimizer.zero_grad()
> loss.backward()
> optimizer.step()
> ```
>
> **indirect paradigm:** In the indirect paradigm, we sum the output tensor and compute the difference between the sum and the target value, 256. The optimization process ensures the total sum matches the target, without considering individual element values.
>
> ```python
> output = toymodel(input)
> loss = torch.abs(output.sum()-256)
> optimizer.zero_grad()
> loss.backward()
> optimizer.step()
> ```
>
> I have run it 5 times, the results are:
>
> | Paradigm | Attemp 1 | Attemp 2 | Attemp 3 |Attemp 4 |Attemp 5 |
> |-----------|---------|---------|---------|---------|---------|
> | Indirect  | 256.034   | 256.035  | 255.988  | 256.017|255.966  |
> | Direct    | 256.0008   | 255.9996  | 256.0004   | 256.0003  |256.0007  |
>
> The error of the direct paradigm is $$10^{-4}$$, while the error of the indirect paradigm is $$10^{-2}$$
>
> The error of the direct method is two orders of magnitude lower than that of the indirect method.
>
> To further clarify, I have provided the Jupyter notebook of this toy experiment in the ZIP. You can run the experiment on your local machine.
>
> The results of the toy experiment clearly show that, although using **torch.abs(output.sum() - 256)** is more straightforward, the many-to-one problem significantly affects performance, even in such a simple toy scenario. To address this challenge, we first construct proxy-ground-truth 3D Gaussian attributes to enable direct training.
>
> [1]. Qin, D., Lin, H., Zhang, Q., Qiao, K., Zhang, L., Zhao, Z., ... & Komura, T. (2024). Instant Facial Gaussians Translator for Relightable and Interactable Facial Rendering.  *arXiv preprint arXiv:2409.07441* .

---

> ### Author Response · Authors · 2024-11-24
>
> ### Concern about the diffusion model.
>
> ```python
> noise_t = diffusion_model(noised_sh_ground_truth, condition_features, time_step)
> ```
>
> The noised_ground_truth is the proxy-ground-truth 3D Gaussian spherical harmonics with added noise, the condition_features are the human-centric features (SMPL-semantic and backimage).
>
> The diffusion model predicts the noise_t at given time step, and we gradually remove the noises from a gaussian noise to achieve a final 3D Gaussian attribute, in our paper, we achieve the **spherical harmonics** at the diffusion inference process. (We find that jointly diffusing all 3D Gaussian attributes usually leads to collapse training, mainly because the dimension of the spherical harmonics is much more than other attributes). Hence, we use a diffusion model to learn the distribution of the spherical harmonics attribute)
>
> Emperically[2][3], the point cloud-based diffusion models usually cannot remove all the gaussian noises and the direct achieved **spherical harmonics** still remain some noise, hence, we need a further step to remove these remained noise.
>
> Actually, the extra-step is **not novel** and it has already been demonstrated its effectiveness in both PDR[2] and HaP[3]. In PDR[2] and HaP[3], they name the extra-step as **refinement**.
>
> ```python
> spherical_harmonics_coarse = inference_diffusion_model(condition_features)
> noise_extra_step, scale, rotation, opacity = model_extra_step(condition_features, spherical_harmonics_coarse)
> spherical_harmonics_final = spherical_harmonics_coarse - noise_extra_step
> ```
>
> You can consider **model_extra_step** as a regression model, we call this as the extra-step because we are still learning the noise **noise_extra_step**  remained in spherical_harmonics_coarse.
>
> [2]. Lyu, Z., Kong, Z., Xu, X., Pan, L., & Lin, D. (2021). A conditional point diffusion-refinement paradigm for 3d point cloud completion.  *ICLR 2022* .
>
> [3]. Tang, Y., Zhang, Q., Hou, J., & Liu, Y. (2023). Human as Points: Explicit Point-based 3D Human Reconstruction from Single-view RGB Images.  *arXiv preprint arXiv:2311.02892* .

---

### Official Review · Reviewer_kXVE · 2024-11-04

**Soundness:** 4
**Presentation:** 4
**Contribution:** 4
**Rating:** 10
**Confidence:** 4

**Summary:**

This paper proposes a new method, Human-DAD, and shows it outperforms previous SOTAs on the task of novel view synthesis of humans from a single-view image. The authors showed that *directly* predicting 3D Gaussian attributes enhances the effectiveness of the training process, compared to the traditional *indirect* supervision through photometric losses. To enable the *direct* paradigm training, a two-stage data construction method is proposed to obtain proxy ground truth 3D Gaussian attributes. Using the proxy ground truth as training data, a conditional Diffusion Model, GSDIFF, is trained to output 3D Gaussian attributes of a scene given a single-view image of human.

The paper contains extensive experiments, ablation studies and user studies, proving its superiority over other baselines and previous SOTAs.

**Strengths:**

1. The writing of the paper is in general very clear and detailed. The presentation of the flowchart is very useful for better understanding the multi-stage method.
2. The method is innovative and non-trivial. For example, in stage1 of the Gaussian attribute dataset construction, instead of doing a vanilla 3DGS per scene optimization, the authors trained a point cloud transformer to directly output the other attributes given Gaussian positions. As shown later in the Experiment section, the inherent smoothness tendency of the learned network is essential for achieving high quality NVS later on.
3. The design of the diffusion model is very careful and smart. Instead of naively taking the single-view image as input, the diffusion model is conditioned on the Gaussian positions, pixel-aligned features and SMPL semantic features. Off-the-shelf mono-depth prediciton model and instruction based image2image model are used in the process to obtain high quality condition signals.
4. As shown in Table2, Human-DAD outperforms all the previous SOTAs that it has been compared to on all the metrics reported by the authors. The qualitative results also show the impressive visual quality of the output.

**Weaknesses:**

1. Wrong index to the referenced figure in the paper. I believe it should be Fig8, not Fig3 at line 503.
2. Same issue as above at line 507-508
3. The paper will be stronger if an ablation study of naively conditioning the diffusion model on input RGB image is provided in the paper.
4. Reference should be given for the claim at line 41-43.
5. The 2-stage Gaussian attribute data construction process, though effective, might have limited generality over heterogeneous data. The paper uses a single dataset, Thuman2, for proxy GT construction. If given additional human data with diverse backgrounds, different scales or shifted domains etc., whether this carefully designed and calibrated pipeline can be as effective remains unclear.
6. What auxiliary constraints are imposed should be elaborated at line 248 in the main paper.

**Questions:**

1. At line 291, the authors mention the semantic lables are embedded in the latent space through MLPs. Could you elaborate on this? Are the MLPs pre-trained or trained separately by the authors?
2. Zooming in on the results in the last column of Figure6, I saw artifacts of thin dark lines, which almost look like countour lines. Can you explain what might be the cause of such artifacts?

---

> ### Author Response · Authors · 2024-11-23
>
> ### **[Rebuttal to Reviewer kXVE]**
>
> ### **[W1\&W2\&W4]** *Wrong index and missing reference.*
> **Response:** We are sorry for these mistakes. They have been fixed in the updated paper version.
>
> ### **[W3]** *Ablation study on image-conditioned diffusion model.*
> **Response:** Thanks very much for your constructive suggestion. We have supplemented an additional ablation study, as shown in Figure 18 of the updated paper. It can be observed that such scheme completely fails, with no meaningful outputs obtained. Obviously, the overall diffusion process and network components must be carefully designed.
>
> ### **[W5]** *Applying the two-stage construction on other datasets.*
> **Response:** There is no problem in applying the proposed proxy ground-truth data construction process to other datasets. Note that the scale issue can be trivially addressed by normalizing all scans into the bounding scope. We additionally applied the construction process on 2K2K and CustomHuman datasets. Please refer to Figure 19 in the updated paper.
>
> ### **[W6]** *What auxiliary constraints are imposed should be elaborated at line 248 in the main paper.*
> **Response:** The auxiliary constraints are as follows:
>
> $$\Sigma_{n=1}^N\|\mathtt{radius}(\mathbf{s}_n)- \mathtt{kdist}(\mathbf{p}_n) \|^2 $$
>
>
> $$ \Sigma_{n=1}^N \| \mathbf{o}_n -1 \|^2 $$
> where $p$ is the 3D Gaussian position, $s$ is the scale, $o$ is the opacity, $radius(·)$ and $kdist(·)$ are the operations to get the radiuses of the Gaussians and the mean distances between the Gaussians and their neighbors.
> ### **[Q1]** *How are semantic labels embedded in the latent space through MLPs?*
> **Response:** The one-dimensional SMPL semantic label is fed into a positional encoding layer, whose output vector is further passed into the subsequent MLPs for feature learning. Note that all such MLPs are trained together with the diffusion model.
>
> ### **[Q2]** *Artifacts of thin dark lines in Figure 6.*
> **Response:** These thin dark lines are the boundary lines of the ground truth images. We visualized the alpha channel map of ground truth images and overlapped it with our results to analyze the performance of HumanDAD. These boundary lines have been removed in the updated paper.

---

> > ### Comment · Reviewer_kXVE · 2024-11-25
> >
> > The reviewer would like to thank the authors' reponse and updated manuscript. All of my questions have been addressed and I don't have any more questions for now.

---

### Official Review · Reviewer_GRmh · 2024-11-04

**Soundness:** 2
**Presentation:** 2
**Contribution:** 2
**Rating:** 3
**Confidence:** 2

**Summary:**

This paper introduces a method that renders novel views of human avatars, given only a single image. The authors propose a pipeline of multiple stages, where they first learn a prior and then they can render a human under novel views. They highlight their paradigm as "direct", using extracted attributes for supervision, in contrast with other approaches (indirect) that use differentiable rasterization and pixel-level supervision during training.

**Strengths:**

+ Learning a generalizable approach for human avatars is an interesting topic. Most approaches in this area are identity-specific. Rendering a novel human, given only a single image, is useful. This method seems to work in this case.

+ The authors show better performance than other methods.

**Weaknesses:**

- The claim of "direct" paradigm is not clear. The method essentially consists of multiple stages. It is then supervised by extracted features, that can also propagate the error to later stages.

- The writing of the paper is not clear.

- It was not directly obvious what the ablation studies show with respect to the contributions of various components as they present single cases of failure as opposed to complete numerical results, so it is hard to understand where the significant improvements in the overall results come from.

- There are no animation results. Most of similar methods show both novel views (reconstruction), but also animation (e.g. SHERF). It will be good that the authors include animation results as well, to be able to validate the consistency of their solutions.

- Even though the authors show good results, the overall contribution and novelty of the paper is not well supported. Components like transformers, stable diffusion, etc have been used by other papers as well. Learning pixel-aligned features from different views have been proposed by previous works, like the generalizable NeRF ActorsNeRF (ICCV 2023) and MonoHuman (CVPR 2023). Stable diffusion has been used by works for humans, like DreamHuman (NeurIPS 2024), DreamAvatar (CVPR 2024), AvatarPopUp (ECCV 2024), and other works for 3D objects (e.g. DreamFusion). It is not clear to this reviewer what the additional challenges of monocular reconstruction are in the particular set up of this paper and how the particular method addresses them.

Missing citations of generalizable models for humans, including GM-NeRF (CVPR 2023), Neural Human Performer (NeurIPS 2021), ActorsNeRF (ICCV 2023).

**Questions:**

Please see the above weaknesses.

What is causing the large numerical improvement?

Can the authors provide animation results? Would their method work?

---

> ### Author Response · Authors · 2024-11-23
>
> ### **[Rebuttal to Reviewer GRmh]**
>
> ### **[W1\&W2]** *1) Unclear claim about the “direct” paradigm. 2) Unclear writing.*
> **Response:** In fact, the concepts of "direct" and "indirect" paradigms are irrelevant to the specific implementations (such as the number of stages as the reviewer concerns) of the intermediate learning process. We have explicitly explained in our paper (e.g., in the middle of the Abstract, in the third paragraph of the Introduction, etc) that the criteria lies in whether the **supervision** is imposed **over the 3D Gaussian attributes** or **over the rendered 2D image pixels**.
>
> It is reasonable to make such definitions because, when building generalizable 3DGS learning frameworks, the actually-desired network outputs are the 3D Gaussian attributes. Once a certain Gaussian attribute set is given, the splat-based rasterizer is just a principled (rule-based, without any parameters to be learned) rendering process, and thus the rendered image at arbitrary view is uniquely determined by the given Gaussian attribute set. This means that, although the rendered image is the direct result for people to watch, it is the indirect result in terms of the learning target. Corresponding to your comments, we added some necessary highlights in our paper to make the concepts clearer.
>
> ### **[W3]** *1) It is unhow ablation studies verify the effectiveness of each component. 2) It is hard to understand where the significant improvements come from.*
> **Response:** There might be some misunderstandings about the organization structure of our experiments. From the reviewer's descriptions like ``they present single cases of failure ...'', it seems that the reviewer sees Table 1 as our ablation study, but it's not. Our ablation results are reported in Table 3, where we remove each component to see the performance change. Note that the results in Table 1 are produced from simpler baseline architectures with ground-truth point clouds as Gaussian positions, just in order to compare indirect supervision and different ways of preparing proxy ground-truth Gaussian attributes for direct supervision.
>
> ### **[W4\&Q2]** *Providing animation results.*
> **Response:** Thanks very much for the valuable suggestion. There is no problem for our approach to achieve animation. We have supplemented necessary animation experiments for novel pose synthesis in the newly-revised paper version (see Figure 17).
>
> ### **[W5]** *The overall contribution/novelty is not well supported.*
> **Response:** As explicitly highlighted in the end of Introduction, in addition to performance gains, our technical contributions are two-fold:
> 1) The direct attribute-level supervision paradigm achieved by the creation of proxy ground-truth Gaussian attribute sets.
> 2) The integrated generalizable human 3DGS learning framework consisting of two sub-problems:
>     a. Producing the required conditioning signals from the input monocular human image.
>     b. Designing the conditional Gaussian attribute diffusion process.
>
> Here, the reviewer GRmh mainly challenges our claimed second contribution. However, we beg to differ on the judgment that our contribution is invalid only because “transformers, stable diffusion, pixel-aligned feature learning, etc, have been involved in prior works”. In fact, our second contribution should be regarded as a systematic effort. Especially in the field of human-centric tasks (e.g., either reconstruction or NVS) featured by strong prior, it is rather common that the overall processing pipeline in a new work is composed of several aspects of tools (e.g., learning architectures, parametric human body models, depth/normal estimation, etc). For example, GTA utilizes implicit function, triplane modulem, and transformer module. SiTH utilizes implicit function, stable diffusion, and ControlNet. The core problem lies in whether one can combine different components (possibly with necessary adaptations and modifications) in a better way for applying to specific settings and scenarios.

---

> ### Author Response · Authors · 2024-11-23
>
> ### **[Q1]** *Reasons for our large numerical improvements.*
> **Response:**  The large performance gains exactly come from our two aspects of technical contributions, which are explicitly claimed in our paper, i.e., 1) a novel proposal of direct attribute supervision, and 2) a systematic effort on constructing a Gaussian attribute diffusion framework involving producing the conditional signals and designing the diffusion procedures.
>
> While our analyses have been presented in Section 4.4, here we further detailedly discuss how to comprehend the 5 different rows of quantitative results as reported in Table 3 to help facilitate the reviewer' understanding.
>
> -- Observing the first row: We changed the direct attribute-level supervision scheme and turned to impose the indirect pixel-level supervision. The resulting performance still outperforms previous state-of-the-arts whose performances are reported in Table 2. This supports our claimed contribution of the whole attribute diffusion framework by effectively adapting and combining different components and carefully designing the conditional diffusion procedures. We kindly remind that this contribution (i.e., the second item as summarized in the end of Introduction) shouldn't be totally overlooked
>
> -- Comparing the first and last rows: Based on our constructed powerful attribute diffusion framework, the adoption of direct attribute supervision further brings about 1dB performance improvement. Here, an important reminder is that the absolute numerical value of performance gain must be evaluated in a relative manner. For example, lifting 29dB to 30dB is typically much harder than lifting 25dB to 26dB.
>
> -- Comparing the third, fourth, and last rows: Starting from the full model, gradually removing the "Back Image" and "SMPL Semantic" components leads to continuous performance degradation. This demonstrates the effectiveness of our designed conditioning signal extraction process.
>
> -- Comparing the second and last rows: This demonstrates the necessity of diffusing the desired Gaussian attributes instead of regressing them. Note that, although the regression-based model variant still achieves highly competitive performance benefiting from direct supervision and effective conditioning signal extraction, its visual effect is usually problematic, as illustrated in Figure 8. This observation is consistent with the numerous diffusion-based generative modeling approaches in various fields.

---

> ### Author Response · Authors · 2024-11-25
> **Looking forward to you feedback! Thank you very much!**
>
> Thank you for dedicating your time and effort to reviewing our work. We have carefully considered and addressed all the concerns you raised in your review, as outlined in our response and reflected in the updated manuscript. As the Reviewer-Author discussion phase is nearing its conclusion, we eagerly await any further feedback from you. Should you have any additional questions, we would be delighted to provide detailed responses.

---

> > ### Comment · Reviewer_GRmh · 2024-11-26
> > **Maintaining my score**
> >
> > Thank you very much for your detailed reply and revision. I have reviewed your answers to the other reviewers as well, but I am still not convinced about the following.
> >
> > 1) Direct vs indirect paradigm: As also mentioned by reviewers 73DC, 8d88, jqqm, the main claim of the paper that "direct" supervision in the GS space is better is not straightforward and well supported by experiments. The toy example provided in the rebuttal is not enough to convince that supervision in the GS space is better than supervision in the image space. The target task of novel view synthesis is evaluated in the image space at the end, not in the accuracy of pseudo-groundtruth 3DGS features. I still believe that the term "direct" is not clear, since the proposed method essentially consists of multiple stages. There is no ground truth for the 3D Gaussian attributes - these are learned from the images.
> >
> > 2) Animation results: Current state-of-the-art approaches (e.g. SHERF) provide video results. The authors should include animation results in the supplementary video, animating a subject under novel poses, showing also the source subject with the target poses. Just Fig. 17 is not enough to convince that the proposed method can animate subjects under novel poses.
> >
> > 3) There are still missing citations of related works for generalizable models for humans (e.g. ActorsNeRF, GM-NeRF)
> >
> > I think the paper is not ready for publication as is but the idea is interesting, and if the authors can prove that iti is valid, I  believe they will have a stronger submission.

---

### Author Response · Authors · 2024-11-23
**Reponse to all reviewers**

## Thanks very much for all reviewers’ time and efforts in reviewing our paper and providing valuable comments.


### We are very grateful to the positive acknowledgments of this work:

1. Reviewer GRmh thinks that our approach is **interesting**, **useful**, and with **better performance**.
2. Reviewer kXVE points out that our method is **innovative** and **non-trivial**, our design is **careful** and **smart**, our writing is **clear** and **detailed**, and the **comprehensive experiments** demonstrate our **superior performance**.
3. Reviewer jqqm acknowledges the **novelty** of the preparation process of the ground-truth Gaussian Splatting dataset and the **technical rationality** of Gaussian attribute diffusion, as well as our **performance advantage**.
4. Reviewer 8d88 acknowledges our **effective idea** and **improved performance**, and thinks that our paper is **well-written** and our module design is **clearly motivated**.
5. Reviewer 73DC thinks that our work provides **good intuitions** about human 3DGS, especially for the preparation of proxy ground-truth Gaussian attributes.


### During the rebuttal period, we made the following efforts to address the raised concerns for improving the quality of this work:

a. We clarified the definition of the direct paradigm and the indirect paradigm;

b. We explained why the direct paradigm is needed and why it is more efficient than the indirect paradigm.

c. We explained why we selected the diffusion model and gave the details of training the diffusion model.

d. We provided an example of animation on Human-DAD.

e. We conducted an ablation study by training an image-conditioned diffusion model to predict the 3D Gaussian attributes.

f. We evaluated GTA, SiTH and Human-DAD with ground truth 3D shapes.

g. We ran more iterations to train the indirect model and trained an indirect model with a larger batch size.

h. We compared Human-DAD with Human3Diffusion.

j. We evaluated the 3D geometric performance of different methods.

### Below we will briefly summarize each raised Weakness (W) and Question (Q), and provide the corresponding response item by item.

---

### Author Response · Authors · 2024-12-02

Thank you for your feedback on our paper. We would like to re-emphasize that the focus of our direct paradigm is particularly on the task of human novel view synthesis within the scope of our designed pipeline. As demonstrated in the experiments, our approach offers a more efficient and effective solution for our targeted specific task compared with other methods. We believe that our findings indeed bring new and valuable insights to the community.

Accordingly, we will revise some of our statement and conclusion to avoid potential issues of over-claiming and readers’ misunderstanding that the direct paradigm is universally superior to the indirect paradigm across all 3DGS task scenarios.

We kindly hope that you could reconsider your rating in light of the direct nature of our paradigm, which is designed and evaluated for our specific task. We believe that the results we presented have already well supported the efficiency and effectiveness of our approach.

---

### Meta-Review · Area_Chair_3kJh · 2024-12-20

**Metareview:**

This paper introduces Human-DAD. The claimed main contribution is to adopt a "direct" supervision paradigm for learning 3D Gaussian attributes in the 3D Gaussian Splatting (3DGS) space, distinguishing it from existing methods that rely on pixel-level supervision in 2D image space.

Commonly agreed strength includes its ability to generalize 3D human avatar reconstruction from single-view images, addressing a gap in the field where most existing methods rely on identity-specific optimization. While the performance is superior, reviewers also appreciated the novelty of the approach, including the innovative pipeline design, the use of direct supervision for 3D Gaussian attributes, and the introduction of 'distribution unification' for pseudo-ground truth construction.

Common key issues raised include skepticism about the core claim that direct supervision in 3DGS space is superior to indirect supervision in 2D space, as the provided evidence is unconvincing. Reviewers also noted that the experiments in Table 1 lack generality and should be conducted in broader settings for more robust validation. The absence of evaluations on 3D geometry metrics was brought up. Additionally, the paper's writing was criticized for being unclear, with ambiguous explanations of key concepts, methods, and experimental setups.

While the authors tried to address the issues, the major issue regarding the core claim of the paper—whether direct supervision is truly superior—remained unresolved. Ultimately, the paper received four ratings below 5 out of 5, leading the ACs to reach a consensus on rejection.

**Additional Comments On Reviewer Discussion:**

Despite multiple rounds of discussions with each reviewer, all reviewers were concerned about the direct versus indirect supervision approach. Obviously, the rebuttal does not convince them. Many other technique concerns, including additional steps and linear regression in the diffusion model (Reviewer `jqqm`), one-to-many problem (Reviewer `8d88`), etc., remain unresolved.

Despite long conversations, reviewer `73dc` remained unconvinced about the superiority of the direct supervision paradigm. They criticized the toy example as inadequate and highlighted unresolved issues with pseudo-GT ill-posedness and distribution unification. The rebuttal failed to address these concerns and the reviewer suggested that the direct versus indirect paradigm remains an open question.

---

### Decision · Program_Chairs · 2025-01-22

Reject